# Rollout Total Correlation for Deep Reinforcement Learning

**Bang You**  *bangyou@whut.edu.cn*
*School of Information Engineering*
*Wuhan University of Technology*

**Huaping Liu**[*]  *hpliu@tsinghua.edu.cn*
*Department of Computer Science and Technology*
*Tsinghua University*

**Jan Peters**  *jan.peters@tu-darmstadt.de*
*Intelligent Autonomous Systems*
*Technische Universität Darmstadt*
*German Research Center for AI (DFKI)*
*Hessian Centre for Artificial Intelligence (Hessian.AI)*
*Centre for Cognitive Science (CogSci)*

**Oleg Arenz**  *oleg.arenz@tu-darmstadt.de*
*Intelligent Autonomous Systems*
*Technische Universität Darmstadt*

**Reviewed on OpenReview:** *https://openreview.net/forum?id=qTdRJAL8Li*

## Abstract

Learning task-relevant representations is crucial for reinforcement learning. Recent approaches aim to learn such representations by improving the temporal consistency in the observed transitions. However, they only consider individual transitions and can fail to achieve long-term consistency. Instead, we argue that capturing aspects of the state that correlate with other states and actions of the trajectory—even more distant in the future—could further help in extracting task-relevant information. Hence, in this paper we investigate how to learn representations by maximizing the rollout total correlation, the correlation among all learned representations and actions within the trajectories produced by the agent. For improving rollout total correlation, we propose to combine two complementary lower bounds based on a generative and a discriminative model, combined with a simple and effective technique of chunk-wise mini-batching. Furthermore, we propose an intrinsic reward based on the learned representation for better exploration. Experimental evaluations on a set of challenging image-based simulated control tasks show that our method achieves better sample efficiency, and robustness to both white noise and natural video backgrounds compared to leading baselines.

## 1 Introduction

Reinforcement learning (RL) has achieved impressive success by enabling robots to acquire skills directly from high-dimensional images (Laskin et al., 2020). A critical ingredient in reinforcement learning from images is to learn concise representations of high-dimensional images that are relevant to the task at hand. Reinforcement learning agents can subsequently learn policies or Q-functions based on obtained task-relevant representations. However, extracting such representations is challenging, since reinforcement learning objectives have no direct incentive to filter out redundant or irrelevant information in image observations, such as noise or changes in the background. Furthermore, representation learning in RL is deeply intertwined with

---

[*]Corresponding Author: Huaping Liu

the problem of exploration (Yarats et al., 2021a; Yuan et al., 2023): learning representations requires diverse observed data of the agent, while effective exploration can only be achieved with expressive representations.

Previous methods alleviate the above challenges by constructing both auxiliary representation learning objectives and reward functions. Particularly, extracting task-relevant representations of observations can be achieved by constructing auxiliary representation learning objectives that capture temporal consistency in collected transitions (Pathak et al., 2017; Sekar et al., 2020; Guo et al., 2022; Chiappa et al., 2023). An effective technique to capture the temporal consistency is to preserve the mutual information related to environment dynamics (Kim et al., 2019a; Tao et al., 2020; Bai et al., 2021), such as the mutual information between the representation of the next state, and the representation of the current state and the action. However, these prior methods based on mutual-information usually only capture predictive information in individual transitions and may fail to model temporal coherency within sequential observations.

We argue that enforcing temporal consistency throughout longer sequences of states and actions is more efficient for learning task-relevant representations. For example, the energy of an approximately closed system is a very powerful feature, precisely due to its temporal consistency, which allows for long-term predictions. By maximizing the correlation within whole trajectories, we aim to extract such important representations from high-dimensional inputs.

Hence, in this paper we propose the rollout total correlation (ROTOC) objective to learn task-relevant representations that capture temporal consistency in sequences of states and actions (see Fig. 1). Our rollout total correlation objective maximizes the total correlation among latent embeddings and actions in the trajectories encountered by the agent. The captured total correlation can be naturally interpreted as the overall information gain that is related to the temporal evolution of the environment, allowing us to effectively enforce temporal consistency on sequences of the obtained embeddings. We show that the total correlation objective can be lower-bounded by a sum of per-step mutual information objectives that are similar to previous objectives. While this lower bound is more convenient for opti-

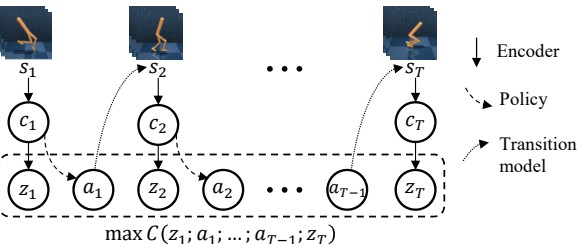

Figure 1: Our rollout total correlation objective maximizes the overall interdependencies between latent state embeddings $\boldsymbol{Z}$ and actions in a sequence for learning representations, which are temporally consistent throughout the sequence.

mization than the total correlation, it misses the long-term correlations that we want to capture. Still, we provide empirical evidence that suggests that representation learning based on such per-step objectives can be biased towards capturing long-term correlation, using a simple technique that we refer to as "chunk-wise mini-batching"–ensuring that the mini-batches contain several subsequent transitions. Our experiments show that this sampling strategy not only increases the long-term predictability of the learned representations, but also increases the performance of the reinforcement learning agent. Furthermore, we encourage the agent to explore novel states by constructing an auxiliary reward that measures the task-relevant state novelty by a discriminative bound on the predictive information.

The main contributions of our work are summarized as follows. We introduce the rollout total correlation objective for representation learning in RL, which aims to learn representations that capture long-term temporal correlations within the trajectory. We simplify our objective into a sum of per-step mutual information, and derive a tractable lower bound of the rollout total correlation, by combining two different lower-bounds of the per-step mutual information. We propose chunk-based mini-batching, a technique for increasing multi-step correlation of the learned embeddings. We propose an intrinsic reward based on our learned representations that quantifies the task-relevant state novelty for guiding exploration. We conduct extensive experiments on a set of challenging image-based continuous control tasks, including standard Mujoco and Mujoco tasks with distractive backgrounds, where we use both, white noise and natural videos. Empirical evaluations show that our method achieves better sample efficiency on high-dimensional Mujoco tasks, while being more robust to observation perturbations than leading baselines.

The remainder of the paper is organized as follows. In Section 2, we discuss prior related work. In Section 4, we present the proposed rollout total correlation objective. We present empirical evaluations in Section 5. In Section 6, we draw a conclusion and discuss limitations and future work.

## 2 Related Work

Unsupervised representation learning methods in RL focus on extracting expressive state representations based on which the agent makes decisions (Yarats et al., 2021b; Zhu et al., 2022; You & Liu, 2024; Tomar et al., 2024; Ni et al., 2024). Common techniques proposed to learn representations include using bisimulation metrics (Kemertas & Jepson, 2022; Zhang et al., 2020; Castro, 2020), reconstructing raw observations (Yarats et al., 2021b; Voelcker et al., 2024), and contrastive learning (Laskin et al., 2020; You et al., 2022). Particularly, task-relevant representations can be extracted from observations by capturing the temporal consistency in collected transitions. An effective technique to capture temporal consistency in transitions is to construct forward or inverse transition models of the environment in the latent space of representations (Sekar et al., 2020; Guo et al., 2022; Zhao et al., 2023). Our work is more related to another group, where capturing the temporal consistency is based on preserving the mutual information related to environment dynamics (Lee et al., 2020; You et al., 2022; Bai et al., 2021; Lee et al., 2023), such as maximizing the mutual information between the representation of the next state, and the representation of the current state and the action. However, prior methods based on mutual information only capture the temporal consistency in individual transitions involving two successive states given the action. Instead, we aim to capture temporal consistency throughout a trajectory. To this end, we derive our method based on a total correlation objective, propose chunk-based mini-batching for maximizing the total correlation, and analyse the relation between long-term predictability and downstream performance of the RL agent.

Some recent works, DRIBO (Fan & Li, 2022) and RPC (Eysenbach et al., 2021), extend the mutual-information objective to the sequential setting in RL for learning task-relevant representations. Fan et al. (Fan & Li, 2022) maximize the task-relevant information shared between representations of the observations from different views and meanwhile filter out the task-irrelevant information not shared by the multi-view observations by using a multi-view information bottleneck loss. While DRIBO also uses sequential data points during training, its objective is not related to the total correlation among embeddings and actions. Eysenbach et al. (2021) maximize external rewards while minimizing the mutual information between a sequence of states and a sequence of its latent representations used by the policy, for limiting the amount of observed information that affects the agent's behavior. RPC introduces an upper bound on the mutual information and use it to construct a new algorithm, which uses a common objective to optimize the policy and the encoder. While our KL-based lower bound on the total correlation takes a similar form to the upper bound used in RPC, we derived it from a different perspective. Furthermore, we combined it with a lower bound based on a discriminative model, introduced and analyzed the simple yet powerful technique of chunk-wise mini-batching, proposed an auxiliary reward for better exploration based on our total correlation bound, and optimized the encoder and the policy using separate objectives.

Our work is also related to exploration with intrinsic motivation in RL (Yarats et al., 2021a; Huang et al., 2024), since we use an intrinsic reward to incentivize the agent to collect diverse data, which our encoder is based on to induce representations. Some exploration approaches are curiosity-driven, which usually build a latent space of representations that filters out irrelevant information in raw states, and then use the difference between predicted representations and true representations as novelty measurement (Sekar et al., 2020; Guo et al., 2022; Hao et al., 2023). The most related approaches are those that apply the information theory principle to distill task-relevant information from observations and then capture task-relevant state novelty in the obtained representation space for better exploration (Tao et al., 2020; Kim et al., 2019b; Bai et al., 2021). Different from existing (non-sequential) information-theoretic methods (Kim et al., 2019a; Tao et al., 2020; Bai et al., 2021) that only capture temporal coherence in individual states or state representations, our total correlation objective aims to enforce temporal consistency on sequential transitions.

The concept of total correlation was recently applied to reinforcement learning by You et al. (2025), albeit as a novel regularizer, with the aim of biasing the policy toward simpler and more predictable behavior. Instead, our work focuses on representation learning and exploration for reinforcement learning from images.

Furthermore, the work by You et al. (2025) optimizes policy and encoder using a loss based on two generative models for latent state and action predictions, whereas, for representation learning, we focus on the latent state predictability using a generative and discriminative bound, and use separate objectives for policy and encoder. Furthermore, we introduce chunk-wise mini-batching sampling and a novel intrinsic reward based on our discriminative total correlation lower bound.

## 3 Preliminaries and Notation

### 3.1 Problem Statement and Notation

We consider a Markov decision process formulated by the tuple $\mathcal{M} = (\mathcal{S}, \mathcal{A}, P, r, \gamma)$, where $\mathcal{S}$ is the state space, $\mathcal{A}$ is the action space, $P(\boldsymbol{s}_{t+1}|\boldsymbol{s}_t, \boldsymbol{a}_t)$ is the stochastic dynamic model, $r(\boldsymbol{s}, \boldsymbol{a})$ is the reward function and $\gamma \in (0, 1)$ is the discount factor. At each time step, the agent observes the current state $\boldsymbol{s}_t$ and selects its actions $\boldsymbol{a}_t$ based on its stochastic policy $\pi(\boldsymbol{a}_t|\boldsymbol{s}_t)$, and then receives the reward $r(\boldsymbol{s}_t, \boldsymbol{a}_t)$. The RL objective is to maximize the expected cumulative rewards $\mathbb{E}_{\mathcal{T}}\left[\sum_{t=1}^{T} \gamma^t r_t\right]$ where $\mathcal{T} = (\boldsymbol{s}_1, \boldsymbol{a}_1, \boldsymbol{s}_2, \boldsymbol{a}_2, \cdots, \boldsymbol{a}_{T-1}, \boldsymbol{s}_T)$ denotes the agent's trajectory with horizon $T$. We focus on image-based reinforcement learning, where the state space is in the form of images. We do not explicitly consider partial observability of single image observations, but assume that stacking the $j$ most recent images yields a high-dimensional but Markovian state $\boldsymbol{s}_t = (\boldsymbol{o}_t, \boldsymbol{o}_{t-1}, \cdots, \boldsymbol{o}_{t-j+1})$ where $\boldsymbol{o}_t$ is the observed image at time step $t$.

### 3.2 Total Correlation

For multiple random variables $\{x_1, x_2, \cdots, x_n\}$, the total correlation (Watanabe, 1960; Studený & Vejnarová, 1998) among these variable $\mathcal{C}(x_1, x_2, \cdots, x_n)$ is defined as the Kullback-Leibler divergence between the joint distribution of all variables and the product of their marginal distributions $p(x_1)p(x_2)\cdots p(x_n)$,

$$\mathcal{C}(x_1; x_2; \cdots; x_n) = \int_{\mathbf{x}_1, \mathbf{x}_2, \cdots, \mathbf{x}_n} p(x_1, x_2, \cdots, x_n) \log \frac{p(x_1, x_2, \cdots, x_n)}{\prod_{t=1}^{n} p(x_t)} \mathrm{d}x_1 \cdots \mathrm{d}x_n.$$

Total correlation is an effective measure of the total statistical dependence between multiple random variables. However, total correlation suffers from the same problem as mutual information, namely, that directly computing total correlation is computationally intractable in general.

## 4 Rollout Total Correlation

We will now formulate our rollout total correlation objective, and derive a tractable lower bound. We also describe how to generate intrinsic rewards, and how our method can be integrated with a RL algorithm.

### 4.1 The Rollout Total Correlation Objective

We introduce the rollout total correlation objective, ROTOC, for capturing temporal correlation within a sequence of observations and actions. We first map a sequence of high-dimensional states $(\boldsymbol{s}_1, \boldsymbol{s}_2, \cdots, \boldsymbol{s}_T)$ into compact deterministic representations $(\boldsymbol{c}_1, \boldsymbol{c}_2, \cdots, \boldsymbol{c}_T)$ using a deterministic convolutional network. We further use a parameterized stochastic neural network that takes state representations as input and randomly outputs latent samples $\boldsymbol{Z}$. The goal of ROTOC is to find long-term temporally consistent representations $\boldsymbol{Z}$, given a sequence of states $(\boldsymbol{s}_1, \boldsymbol{s}_2, \cdots, \boldsymbol{s}_T)$ and actions $(\boldsymbol{a}_1, \boldsymbol{a}_2, \cdots, \boldsymbol{a}_{T-1})$.

The proposed objective of ROTOC aims to maximize the total correlation among a sequence of stochastic representations and actions,

$$\max \quad \mathcal{C}(\boldsymbol{z}_1; \boldsymbol{a}_1; \cdots; \boldsymbol{a}_{T-1}; \boldsymbol{z}_T) \tag{1}$$

which can be defined as the Kullback-Leibler (KL) divergence between the joint distribution of all embeddings and actions and the product of their marginals,

$$\mathcal{C}(\boldsymbol{z}_1; \boldsymbol{a}_1; \cdots; \boldsymbol{a}_{T-1}; \boldsymbol{z}_T) = \mathbb{E}_{p(\boldsymbol{z}_{1:T}, \boldsymbol{a}_{1:T-1})}\left[\log \frac{p(\boldsymbol{z}_{1:T}, \boldsymbol{a}_{1:T-1})}{\prod_{t=1}^{T} p(\boldsymbol{z}_t) \prod_{t=1}^{T-1} p(\boldsymbol{a}_t)}\right]. \tag{2}$$

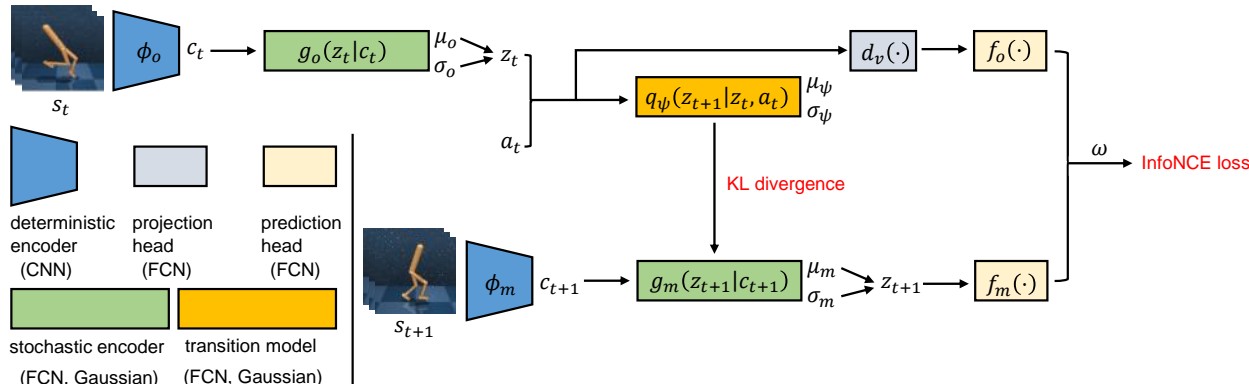

Figure 2: Our network framework. A stack of the most recent image frames is compressed using a deterministic CNN and then fed into a stochastic encoder that we use for sampling latent representations of the states. Our lower bounds on the rollout total correlation objective involve learning a latent transition model with a loss given by a KL divergence, and learning non-linear transformations of $(z_t, a_t)$ and $z_{t+1}$ that are used by the InfoNCE loss. For the deterministic and stochastic encoder ($\phi$ and $g$) and the nonlinear transformations $f$, we use target networks when processing the state at the next time step $t+1$, and online networks when processing the state at the current time step $t$.

The total correlation $\mathcal{C}(z_1; a_1; \cdots; a_{T-1}; z_T)$ qualifies the total interdependencies among a sequence of stochastic representations and actions, which can be intuitively viewed as the additional amount of information that needs to be transferred when encoding every random variable independently instead of using an optimal code for transmitting the whole trajectory. Maximizing the total correlation among state embeddings and actions yields temporally consistent representations that correlate with the agent's actions, and, therefore, focus on task-relevant features.

## 4.2 Algorithm Overview

Unfortunately, maximizing the rollout total correlation directly is computationally intractable, and, hence, we derive a tractable approximation. We use our approximation as an auxiliary loss during reinforcement learning for learning representations. Furthermore, we build an intrinsic reward for guiding exploration.

Before introducing the concrete loss functions, the reward function and their derivations, we provide an overview of the proposed network architecture in Fig. 2. We use an online deterministic encoder $\phi_o$ with parameters $\eta_o$ to extract the state representation $c_t$ from the state $s_t$ at the current time step. An online stochastic encoder with parameters $\theta_o$ is used to map the state representation $c_t$ into mean $\mu_o$ and standard deviation $\sigma_o$ of a diagonal Gaussian distribution $g_o(z_t|c_t) = \mathcal{N}(\mu_o, \text{diag}(\sigma_o^2))$, which is used for sampling the stochastic embedding. The sampled stochastic representation $z_t$ and action $a_t$ are fed into a transition model with parameters $\psi$, which outputs the mean $\mu_\psi$ and standard deviation $\sigma_\psi$ of a variational Gaussian distribution $q_\psi(z_{t+1}|z_t, a_t) = \mathcal{N}(\mu_\psi, \text{diag}(\sigma_\psi^2))$ that we introduce for one of our lower bounds. A projection head $d_v$ with parameters $v$ and an online prediction head $f_o$ with parameters $\rho_o$ are used to introduce a nonlinear transformation to the stochastic representation $z_t$ and action $a_t$ that are used for another lower bound, which relates to InfoNCE (Oord et al., 2018). While determining the next representation $c_{t+1}$ and stochastic representation $z_{t+1}$, we use target networks for preventing gradients to flow through the deterministic and stochastic encoders. We also use a target prediction head $f_m$ without gradient flow when introducing a nonlinear transformation of the next representation $z_{t+1}$ for the InfoNCE loss. The parameters of the target networks are not optimized with respect to our objective functions, but follow their online counterparts using an exponential moving average technique (He et al., 2020). The state representations and their corresponding latent stochastic representations are, hence, created using the models,

$$c_t = \phi_o(s_t; \eta_o), \quad c_{t+1} = \phi_m(s_{t+1}; \eta_m), \quad z_t \sim g_o(z_t|c_t; \theta_o), \quad z_{t+1} \sim g_m(z_{t+1}|c_{t+1}; \theta_m). \tag{3}$$

### 4.3 Lower-Bounding the Rollout Total Correlation

We will now construct a variational lower bound on the total correlation $\mathcal{C}(\boldsymbol{z}_1; \boldsymbol{a}_1; \cdots; \boldsymbol{a}_{T-1}; \boldsymbol{z}_T)$ suitable for optimization. We first simplify our total correlation objective into step-wise losses,

$$\mathcal{C}(\boldsymbol{z}_1; \boldsymbol{a}_1; \cdots; \boldsymbol{a}_{T-1}; \boldsymbol{z}_T) \geq \mathbb{E}_{p(\boldsymbol{z}_{1:T}, \boldsymbol{a}_{1:T-1})}\left[\log \frac{p(\boldsymbol{z}_1)\prod_{t=1}^{T-1} p(\boldsymbol{z}_{t+1}|\boldsymbol{z}_t, \boldsymbol{a}_t)}{p(\boldsymbol{z}_1)\prod_{t=1}^{T-1} p(\boldsymbol{z}_{t+1})}\right] = \sum_{t=1}^{T-1} I(\boldsymbol{z}_{t+1}; \boldsymbol{z}_t, \boldsymbol{a}_t). \tag{4}$$

Please refer to Appendix A.1 for the derivation. Eq. 4 shows that the total correlation can be lower-bounded by the sum of the per-step predictive information $I(\boldsymbol{z}_{t+1}; \boldsymbol{z}_t, \boldsymbol{a}_t)$ between the current representation $\boldsymbol{z}_t$ paired with the action $\boldsymbol{a}_t$, and the next representation $\boldsymbol{z}_{t+1}$. Although for computationally tractable optimization we obtained a lower bound in Eq. 4 that does not explicitly capture the long-term correlation, we propose to maximize the lower bound in multiple consecutive transitions for retaining the long-term correlation between representations. We will show in Section 5.3.1, that retaining the predictive information in at least two sequential transitions helps to improve long-term predictability and overall performance.

### 4.4 Two Realizations of the Lower Bound

While the lower bound in Eq. 4 is much simpler compared to the exact total correlation objective, it is still not tractable in computation. We will next show two realizations of the lower bound that use different techniques to bound the per-step mutual information. The first objective is based on a generative transition model and aims to increase the consistency of the representation, while the second objective is based on a discriminative model, focusing on the representativeness of the representation.

#### 4.4.1 Lower Bound Based on a Generative Model

We obtain a lower bound of the per-step mutual information by introducing a parameterized variational distribution $q(\boldsymbol{z}_{t+1}|\boldsymbol{z}_t, \boldsymbol{a}_t)$ to approximate the dynamic model in the latent space $p(\boldsymbol{z}_{t+1}|\boldsymbol{z}_t, \boldsymbol{a}_t)$:

$$I(\boldsymbol{z}_{t+1}; \boldsymbol{z}_t, \boldsymbol{a}_t) \geq -\mathbb{E}_{p(\boldsymbol{c}_{t+1}, \boldsymbol{z}_t, \boldsymbol{a}_t)}\left[\mathbb{D}_{\mathrm{KL}}\Big(p(\boldsymbol{z}_{t+1}|\boldsymbol{c}_{t+1}) \,\|\, q(\boldsymbol{z}_{t+1}|\boldsymbol{z}_t, \boldsymbol{a}_t)\Big)\right]. \tag{5}$$

The detailed derivation is available in Appendix A.2. The obtained lower bound is given in terms of the expected KL divergence between our stochastic encoder and the transition model $q_\psi(\boldsymbol{z}_{t+1}|\boldsymbol{z}_t, \boldsymbol{a}_t)$. Maximizing the bound encourages the transition model to approximate the dynamics of the representation $\boldsymbol{Z}$ to predict the next representation well. The resulting latent representations are easily predicted by our transition model. We compute the KL divergence analytically based on the means and standard deviations of two diagonal Gaussian distributions $g_m(\boldsymbol{z}_{t+1}|\boldsymbol{c}_{t+1})$ and $q_\psi(\boldsymbol{z}_{t+1}|\boldsymbol{z}_t, \boldsymbol{a}_t)$.

#### 4.4.2 Lower Bound Based on a Discriminative Model

Using the KL-based lower bound in Eq. 5 alone could fail to induce meaningful stochastic representations, since for any deterministic embedding, the KL divergence could always be zero for a constant encoding. To alleviate this problem, we also use the InfoNCE lower bound on the local predictive information term $I(\boldsymbol{z}_{t+1}; \boldsymbol{z}_t, \boldsymbol{a}_t)$, which is based on a discriminative model. While it would be possible to only use the InfoNCE lower bound, we hypothesize that the InfoNCE lower bound, which favors discriminative representations is less effective in capturing temporally consistent representations, compared to the KL lower bound, which explicitly uses a transition model to predict the future.

Let $(\boldsymbol{z}_t, \boldsymbol{a}_t, \boldsymbol{z}_{t+1})$ denote samples randomly sampled from the joint distribution $p(\boldsymbol{z}_t, \boldsymbol{a}_t, \boldsymbol{z}_{t+1})$ which we refer to as positive sample pairs, and let $N$ denote a set of negative samples $\boldsymbol{z}_{t+1}^*$ drawn from the marginal distribution $p(\boldsymbol{z}_{t+1})$. Then, the InfoNCE loss $I_\omega(\boldsymbol{z}_{t+1}; \boldsymbol{z}_t, \boldsymbol{a}_t)$ is given as

$$I_\omega(\boldsymbol{z}_{t+1}; \boldsymbol{z}_t, \boldsymbol{a}_t) = -\mathop{\mathbb{E}}_{p, N}\left[\log \frac{h_\omega(\boldsymbol{z}_t, \boldsymbol{a}_t, \boldsymbol{z}_{t+1})}{\sum_{\boldsymbol{z}_{t+1}^* \in N \cup \boldsymbol{z}_{t+1}} h_\omega(\boldsymbol{z}_t, \boldsymbol{a}_t, \boldsymbol{z}_{t+1}^*)}\right] \tag{6}$$

where the expectation is computed over the joint distribution $p(z_t, a_t, z_{t+1})$. The score function $h_\omega(z_t, a_t, z_{t+1})$ transforming feature variables into scalar scores is given by $h_\omega(z_t, a_t, z_{t+1}) = \exp\left(f_o\left(d_v\left(m(z_t, a_t)\right)\right)^\top \omega f_m\left(z_{t+1}\right)\right)$. Here the function $m(z_t, a_t)$ concatenates $z_t$ and $a_t$ and $\omega$ is a learnable weight transformation matrix. The projection head $d_v(\cdot)$ and the online prediction head $f_o(\cdot)$ nonlinearly transform $m(z_t, a_t)$, whereas the target prediction head $f_m$ introduces a nonlinear transformation to $z_{t+1}$. By maximizing the inner product between the nonlinear transformations of $m(z_t, a_t)$ and $z_{t+1}$, the InfoNCE loss with the score function forces our model to learn temporally predictive representations.

## 4.5 The Loss Function and Optimization

By plugging both the KL lower bound in Eq. 5 and the InfoNCE lower bound in Eq. 6 into Eq. 4, we can maximize the rollout total correlation (Eq. 1) by minimizing the loss

$$\underset{\eta_o, \theta_o, \psi, v, \rho_o, \omega}{\arg\min} \quad L = \underset{p(c_{1:T}, z_{1:T}, a_{1:T-1})}{\mathbb{E}} \left[ \sum_{t=1}^{T-1} \alpha \log \frac{g_m(z_{t+1}|c_{t+1})}{q_\psi(z_{t+1}|z_t, a_t)} - \underset{N}{\mathbb{E}} \left[ \log \frac{h_\omega(z_t, a_t, z_{t+1})}{\sum_{z_{t+1}^* \in N \cup z_{t+1}} h_\omega(z_t, a_t, z_{t+1}^*)} \right] \right]. \tag{7}$$

The derived loss contains a summation over time steps and enables us to preserve predictive information in time-series observations. We optimize the loss via sampling during training. The learnable parameters of our ROTOC model are simultaneously optimized by minimizing this loss.

### 4.5.1 Chunk-Wise Mini-Batching

The loss function (Eq. 7) is optimized using stochastic gradient descent based on mini-batches sampled from the replay buffer of the agent. When sampling individual transitions independently, we would only use a single transition for many trajectories, which may prevent us from capturing longer-term correlations. Instead, we propose to sample a minibatch of sequence chunks $(s_t, a_t, s_{t+1})_{t=k}^{L+k-1}$ with chunk length $L$ from the replay buffer. While our loss (Eq. 7) decomposes additively across time steps, implying that chunk-wise sampling does not alter the optimized loss, it intuitively biases the optimization toward local optima with better long-term consistency by ensuring the mini-batch contains subsequent time steps. This is because gradients from independent single-step transitions can steer the model into local optima exhibiting only local temporal consistency, suboptimal for multi-step prediction. Even if subsequent time steps are eventually sampled in other minibatches, escaping such local optima can be difficult. Instead, chunk-wise mini-batching ensures the aggregated gradient of subsequent transitions will agree on pushing towards a globally consistent representation, effectively canceling out signals that might otherwise lead to merely locally consistent solutions. In our experiments, we evaluated the gradient variance across the same fixed test trajectories during training, and observed that the gradient variance along the trajectories is indeed smaller throughout optimization—and most importantly at the end of it, see Appendix D.5.

## 4.6 Intrinsic Reward for Exploration

By minimizing the loss (Eq. 7), we can induce a compact latent space of stochastic representations that filters out distractive information. These representations by themselves can help the agent to more consistently and efficiently explore the environment because filtered information can no longer affect its policy. In this section, we discuss how we can further improve exploration using the task-specific novelty measured in the learned latent space for constructing an intrinsic reward. Our intrinsic reward function is based on curiosity-driven exploration (Guo et al., 2022) where the agent is driven to visit novel states with high prediction errors. More precisely, we provide an additional reward to the agent based on the InfoNCE loss, which measures how difficult it is to predict the next stochastic representation $z_{t+1}$ given the current stochastic representation $z_t$ and the action $a_t$. The intrinsic reward at the current time step is given by

$$r^*(s_t, a_t) = - \underset{p(z_t, a_t, z_{t+1})}{\mathbb{E}} \left[ \underset{N}{\mathbb{E}} \left[ \log \frac{h_\omega(z_t, a_t, z_{t+1})}{\sum_{z_{t+1}^* \in N \cup z_{t+1}} h_\omega(z_t, a_t, z_{t+1}^*)} \right] \right]. \tag{8}$$

Intuitively, the intrinsic rewards encourage the agent to choose the actions that result in transitions that our model is not able to predict well. We use the discriminative model instead of the generative model $q(\boldsymbol{z}_{t+1}|\boldsymbol{z}_t, \boldsymbol{a}_t)$ to compute the intrinsic reward, because discriminative models are typically easier to learn. Combing the environment reward $r(\boldsymbol{s}_t, \boldsymbol{a}_t)$ with the intrinsic reward $r^*(\boldsymbol{s}_t, \boldsymbol{a}_t)$, we obtain the augmented reward $r_{\text{aug}}(\boldsymbol{s}_t, \boldsymbol{a}_t) = r(\boldsymbol{s}_t, \boldsymbol{a}_t) + \lambda r^*(\boldsymbol{s}_t, \boldsymbol{a}_t)$ where $\lambda$ is a scaling factor.

### 4.7 Plugging into a Reinforcement Learning Algorithm

We train our representation jointly with a soft actor-critic agent (SAC) (Haarnoja et al., 2018). We only feed the first transitions of the sampled sequence chunks into the actor and critic for optimizing the agent. Following prior work (Lee et al., 2020; Bai et al., 2021), the actor takes the deterministic state representations $\boldsymbol{c}_t$ as input, which empirically outperforms using the stochastic embeddings $\boldsymbol{z}_t$. We let the gradients of the critic backpropagate through the online deterministic encoder, since the environment reward can provide useful task-relevant information. We stop the gradient of the actor through the embedding, since this can degrade performance by implicitly modifying the Q-function during the actor update (Yarats et al., 2021b). We refer to Algorithm 1 for the training procedure.

## 5 Experimental Evaluation

### 5.1 Experimental Setup

We evaluate the ROTOC on a set of challenging standard Mujoco tasks from the Deepmind control suite (Tassa et al., 2018) (Fig. 3 top row). Specifically, we evaluate our method in six mujoco tasks with high-dimensional observations: Ball-in-cup Catch, Cartpole Swingup-sparse, Reacher Easy, Cartpole Swingup, Walker Walk, and Cheetah Run.

We further carry out our evaluation in two more settings for testing the robustness of our algorithm: the noisy setting and the natural video setting. We refer to Mujoco tasks in the noisy setting and the natural video setting as noisy Mujoco tasks and natural Mujoco tasks, respectively. In the noisy setting, each background image is filled with pixel-wise Gaussian white noise (see Fig. 3 middle row) that is regarded as a strong state distractor for reinforcement learning tasks (Zhou et al., 2023). Furthermore, in order to simulate robots in natural environments with complex observations, in the natural video setting (see Fig. 3 bottom row) the background of the Mujoco tasks is replaced by natural videos (Zhang et al., 2020) sampled from the Kinetics dataset (Kay et al., 2017).

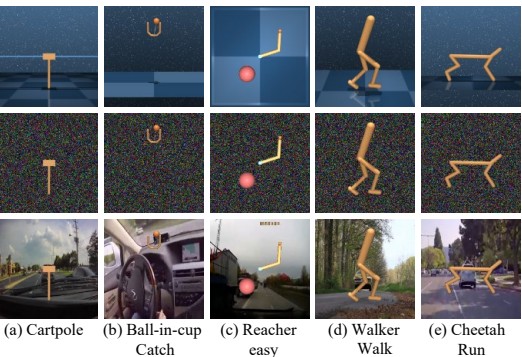

(a) Cartpole (b) Ball-in-cup Catch (c) Reacher easy (d) Walker Walk (e) Cheetah Run

Figure 3: Image-based control tasks used in our experiments. The images show task observations of standard Mujoco tasks (top row), noisy Mujoco tasks (middle) and natural Mujoco tasks (bottom).

The backgrounds of natural Mujoco tasks are continuously changing during training and evaluation, which introduces realistic and strong perturbations to observation images.

We compare ROTOC with the following leading baselines in our experiments: 1) Proto-RL (Yarats et al., 2021a), which extracts prototypical representations from observations and builds an entropy-based intrinsic reward for exploration, 2) DB (Bai et al., 2021), which learns dynamic-relevant compressed representations by using an information bottleneck objective and constructs an intrinsic reward function based on information gain to boost exploration, 3) Plan2Explore (Sekar et al., 2020), which learns task-relevant representations by modeling dynamics and utilizes the disagreement in the predicted next state representations as intrinsic rewards, 4) RPC (Eysenbach et al., 2021), which learns predictable representations from observations and constructs an intrinsic reward function based on the KL divergence, 5) SAC (Haarnoja et al., 2018), which learns policies by maximizing the policy entropy and rewards. We use the same codebase of the SAC algorithm for DB, Proto-RL, RPC, and SAC to ensure a fair comparison to other model-free methods, while following the original implementation of the model-based method Plan2explore. The implementation details

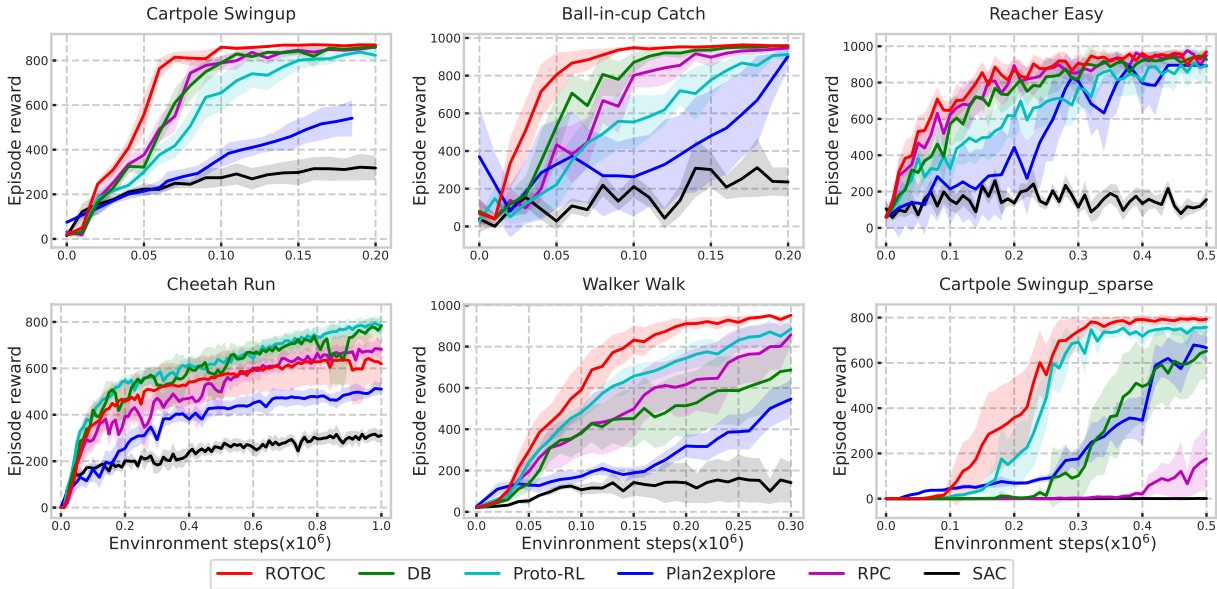

Figure 4: Learning curves of our method and baselines. The plot shows the average reward over 10 seeds and 90% confidence interval. ROTOC outperforms the baselines on the majority of the six standard tasks from the Deepmind control suite in terms of sample efficiency.

of our method and baselines are available in Appendix B. We select the coefficients $\alpha$ and $\lambda$, and the chunk length $L$ by performing hyperparameter tuning on the standard Cartpole Swingup task and subsequently fix them for all other tasks. We refer to Appendix B for more details on determining hyperparameters.

## 5.2 Performance Evaluation

In our first set of experiments we evaluate the final performance and sample efficiency of the different methods in Mujoco tasks in the standard, noisy and natural setting.

### 5.2.1 Standard Setting

We first study whether ROTOC can succeed in challenging Mujoco benchmark tasks when using unperturbed images. In Fig. 4 we compare our algorithm with DB, Proto-RL, Plan2explore, RPC, and SAC on six standard Mujoco tasks from the Deepmind control suite. ROTOC achieves better sample efficiency compared to all baselines on five out of six tasks. The results indicate that the learned representations and the intrinsic reward learned by ROTOC significantly help to learn good policies on challenging image-based tasks.

### 5.2.2 Robustness to Noisy Observations

To study the robustness to white noise, we corrupt the image observations in the Mujoco tasks with Gaussian noise (Fig. 3 middle row), introducing random task-irrelevant patterns to the raw observations. In the top half of Table 1, we compare the performance at a fixed number of environment interactions on the noisy Mujoco tasks. ROTOC achieves competitive asymptotic performance at 500K environment steps compared to the baselines across all tasks. Scores in Table 1 are average rewards and standard errors over 10 seeds. Learning curves and an additional plot and table to compare performance drops caused by the presence of white noise are shown in Appendix C. The performance of our method remains consistent when handling noisy Mujoco tasks, whereas the baselines show a clear decline in performance. For example, ROTOC achieves similar performance in standard and noisy Ball-in-cup Catch and Cartpole Swingup-sparse tasks, while the performance of strong baselines DB, RPC, and Proto-RL clearly decreases due to the noise.

Table 1: Scores (mean and standard error for 10 seeds) achieved by our method and baselines at 500k environment steps on six noisy Mujoco tasks and the natural Mujoco tasks. Our method achieves better or at least comparable performance to baselines.

| 500K step scores | | ROTOC(Ours) | DB | Proto-RL | Plan2Explore | RPC |
|---|---|---|---|---|---|---|
| Noisy Mujoco tasks | Cartpole Swingup | **869± 6** | 851 ± 7 | 846± 5 | 773±28 | 814±13 |
| | Ball-in-cup Catch | **964± 7** | 782 ± 155 | 922± 16 | 892±43 | 928±20 |
| | Reacher Easy | **947± 20** | **939 ± 19** | **869± 63** | **851± 132** | **930± 29** |
| | Cheetah Run | **332± 20** | 265 ± 20 | 268± 13 | 251± 22 | 282±18 |
| | Walker Walk | **935± 11** | 886 ± 30 | **915± 15** | 752±36 | 893±14 |
| | Cartpole Swingup-sparse | **768± 14** | 450 ± 133 | 571± 95 | 680± 51 | 72± 91 |
| Natural Mujoco tasks | Cartpole Swingup | **844± 6** | 803 ± 16 | 748± 28 | 143±28 | 812±10 |
| | Ball-in-cup Catch | **947± 12** | 916 ± 17 | 727± 53 | 133±136 | 838±72 |
| | Reacher Easy | **399± 57** | **327 ± 65** | 257± 39 | **362± 192** | **327 ± 73** |
| | Cheetah Run | **308± 16** | **294 ± 16** | **296± 8** | 193± 32 | **279± 12** |
| | Walker Walk | **886± 15** | 841 ± 31 | **882± 20** | 557±54 | 714±66 |
| | Cartpole Swingup-sparse | **118± 45** | 9 ± 5 | 65± 27 | 0± 0 | 5± 4 |

Table 2: Performance of our model ROTOC for 4 different chunk lengths $L$ on the standard Cartpole Swingup-sparse and Ball-in-cup Catch. The bold font indicates the highest average rewards among all methods. Overall, using larger chunk lengths achieves higher average rewards.

| Scores | L=1 | L=2 | L=3 | L=5 |
|---|---|---|---|---|
| Cartpole Swingup-sparse | 601 ± 96 | 793 ± 10 | **804 ± 10** | 796 ± 12 |
| Ball-in-cup Catch | 953 ± 4 | 957 ± 3 | **965 ± 2** | 963 ± 4 |

Table 3: Ablation of ROTOC over chunk length $L$ and batch size $BS$. The table shows the mean and standard error for 10 seeds. When using the same amount of training samples, e.g. 512, 768, and 1280, using sequential transitions achieves better performance than using individual transitions ($L = 1$).

| Configurations | 512 samples | | 768 samples | | 1280 samples | |
|---|---|---|---|---|---|---|
| | L=1, BS=512 | L=2, BS=256 | L=1, BS=768 | L=3, BS=256 | L=1, BS=1280 | L=5, BS=256 |
| Scores | 707± 25 | **793± 10** | 696± 18 | **804± 10** | 712± 24 | **796± 12** |

### 5.2.3 Robustness to Natural Background

We further investigate the robustness of our method on the natural Mujoco tasks (Fig. 3 bottom row), where the natural backgrounds of the observations are constantly changing. The performance at 500K environment steps is shown in the bottom half of Table 1 and additional plots of the results are provided in Appendix C. Our method achieves better or at least comparable performance across all tasks compared to the baselines. Moreover, the performance of baselines (e.g. Proto-RL and Plan2Explore) declines significantly more than our method on the majority of natural Mujoco tasks. The experimental results in the natural video setting show that ROTOC is more robust to task-irrelevant background than baselines.

### 5.3 Ablation Studies

We performed several evaluations to investigate the effect of chunk-wise mini-batching, the intrinsic reward, and the different lower bounds. We also visualize the learned embeddings to investigate which features are encoded. We perform 5 independent runs for all ablation studies in this section, unless otherwise specified.

### 5.3.1 Chunk-wise Mini-batching

We performed ablations to inspect whether preserving predictive information in sequential transitions improves the performance compared to using individual transitions. In all previous experiments we used the fixed chunk length $L = 2$. In the following experiments we first evaluated the performance of ROTOC when

Table 4: Future predictability of the learned representations with different chunk length $L$ and batch size $BS$. For one-step predictions ($n = 1$), the representations of an encoder that was trained with chunk length $L = 1$ can be better predicted in terms of log-likelihood. However, higher chunk lengths yield representations that are better suited for multi-step predictions ($n \geq 2$).

| Log-likelihood | n=1 | n=2 | n=3 | n=5 |
|---|---|---|---|---|
| L=1,BS=256 | **-78.9 ± 0.6** | -98.3 ± 0.3 | -106.1 ± 0.2 | -114.2 ± 0.4 |
| L=1,BS=512 | **-78.8 ± 0.2** | -97.3 ± 0.6 | -105.6 ± 0.1 | -115.6 ± 0.5 |
| L=2,BS=256 | -80.9 ± 0.4 | **-91.9 ± 0.4** | -99.6 ± -0.1 | -109.3 ± 0.4 |
| L=3,BS=256 | -82.5 ± 0.3 | **-92.5 ± 0.4** | **-98.9 ± 0.1** | **-107.1 ± 0.1** |
| L=5,BS=256 | -118.3 ± 0.4 | -125.4 ± 0.3 | -131.4 ± 0.0 | -139.8 ± 0.8 |

using the same batch size, but different chunk lengths. When the chunk length $L$ is set to 1, ROTOC only uses individual transitions. Using at least two sequential transitions outperforms ROTOC with chunk length $L = 1$ (See Table 2). We also observe that ROTOC using a chunk length of 5 does not further improve performance. We hypothesize that a large chunk length either decreases the variability within the batches (if we compensate for the chunk length by decreasing batch size) or leads to too large batches (if we do not compensate), which may result in convergence to worse local optima (Keskar et al., 2017; LeCun et al., 2002). Furthermore, we investigated the performance of ROTOC when using the same amount of data samples, but different chunk lengths. We ran experiments with $L = 1$ using doubled, tripled, and 5-times larger batch sizes, and compared them to $L = 2$, $L = 3$ and $L = 5$ with a fixed batch-size on the Cartpole Swingup-sparse task. The only difference to our experiment with larger chunk lengths is that the latter guarantees that the minibatch contains consecutive time steps. We can still observe a significant performance benefit of chunk-wise mini-batching (see Table 3), suggesting that the different composition of the mini-batches is the driving factor for the improved performance of ROTOC. These results are in line with our hypothesis that chunk-wise mini-batching can increase total correlation and thereby increases performance.

We further analyze why using sequential transitions improves the performance. To test our hypothesis that using larger chunk lengths increases correlation between more distant time steps, we first learn different encoders using different chunk lengths, and then evaluate the predictability of the learned representations in terms of the log-likelihood loss of a neural network that we train on the respective data sets. The neural network takes the current stochastic or deterministic representations at time step $t$ as inputs and predicts their corresponding embeddings in the future time step $t + n$ with prediction step $n$. Please note, that the different data sets only differ due to the different encoders, as we use the same underlying data set of states $S$, which is taken from the replay buffer of the agent that was trained with chunk length $L = 1$. Table 4 and additional results in Appendix D.1 compare the future predictability of the learned stochastic and deterministic representations with different chunk lengths on the natural Cartpole Swingup-sparse task. Experimental results show that the stochastic and deterministic embeddings learned by using chunk length $L = 2, 3$ are better suited for multi-step prediction, but worse for

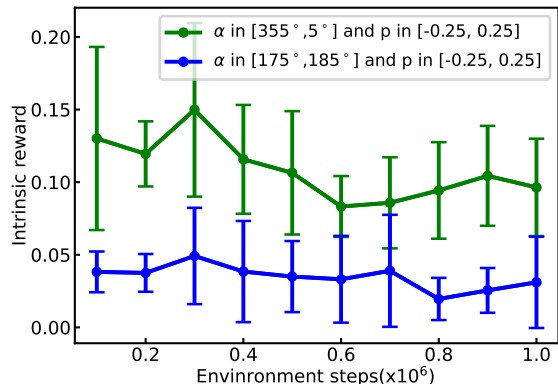

Figure 5: Our intrinsic reward function allocates higher intrinsic rewards to the novel states with position in the range of $[-0.25, 0.25]$ and pole angle in the range of $[355°, 5°]$ compared to the frequently visited states with position in the same range and pole angle in the range of $[175°, 185°]$ on the Cartpole Swingup-sparse task with natural background.

single-step prediction. We attribute the performance gain achieved by using the larger chunk length to improved long-term predictability of the learned embeddings by preserving predictive information in multiple consecutive time steps. With a larger chunk length $L = 5$, the future predictability of the learned

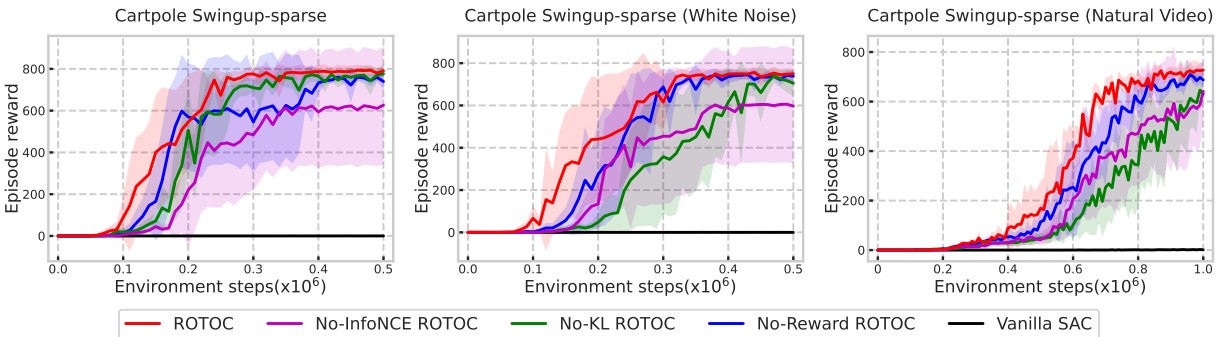

Figure 6: Performance of ablations on Cartpole Swingup-sparse task in the standard setting (left), noisy setting (middle) and natural setting (right).

embeddings degrades compared to using individual transitions. As we discussed above, we hypothesize that too large chunk lengths lead to too small variance in the stochastic gradient estimates, which may hurt the performance.[1]

### 5.3.2 Intrinsic Reward

We also study whether the intrinsic reward function effectively guides the agent to explore novel states in the presence of natural backgrounds. Specifically, on the natural Cartpole Swingup-sparse task, we store snapshots of our intrinsic reward function every 100,000 steps and evaluate the rewards on a fixed data set. We split the data set into two subsets, where one of the subsets only contains particularly relevant transitions and the other subset contains all remaining transitions. We consider states with a pole angle in the range of $[355°, 5°]$, which corresponds to an upright position, as particularly relevant (see more details in Appendix D.4.1). Fig. 5 shows the mean and standard deviation of the intrinsic for the two subsets, each for the same 10 different snapshots of the intrinsic reward function. We also show the evolution of intrinsic rewards along near-optimal trajectories sampled from the fixed data set in Appendix D.4.2. The results show that our intrinsic reward function allocates higher reward values to the novel states than the frequently visited states, indicating that our intrinsic reward can effectively guide exploration in the presence of natural backgrounds.

### 5.3.3 Individual Contributions of Each Component

We analyze the effects of the individual contributions on three ablations: No-Reward ROTOC lacks the intrinsic reward signal, No-KL ROTOC lacks the first terms in Eq. 7, and No-InfoNCE ROTOC lacks the second terms in Eq. 7 and does not use intrinsic rewards. We also evaluate vanilla SAC, which uses the same convolutional encoder and SAC architecture as ROTOC, without any additional representation learning loss, intrinsic rewards, or chunk-wise mini-batching. Fig. 6 shows the results (mean and 90% confidence interval over 5 seeds) on the standard Cartpole Swingup-sparse task with raw images, white noise, and natural video background. Appendix D.2 shows the results of all three sparse-reward tasks. ROTOC achieves better or comparable performance than its ablations on all tasks. In Appendix D.3, we also show the effect of the KL lower bound on the long-term predictability akin to the experiments for Table 4. The results suggest that our lower bound based on a generative predictive model further increases total correlation.

### 5.3.4 Visualization

We visualize our learned representations $Z$ on the natural Mujoco tasks using the Stochastic Neighbor Embedding technique (t-SNE) (Maaten & Hinton, 2008) for projecting the representations into 2D plots. To avoid overlapping, we quantize t-SNE points into a 2D grid of size $20 \times 15$ using RasterFairy (Klingemann,

---

[1]The variance across different stochastic gradient estimates is not to be confused with the variance of the per-time-step contributions within a single stochastic gradient estimate.

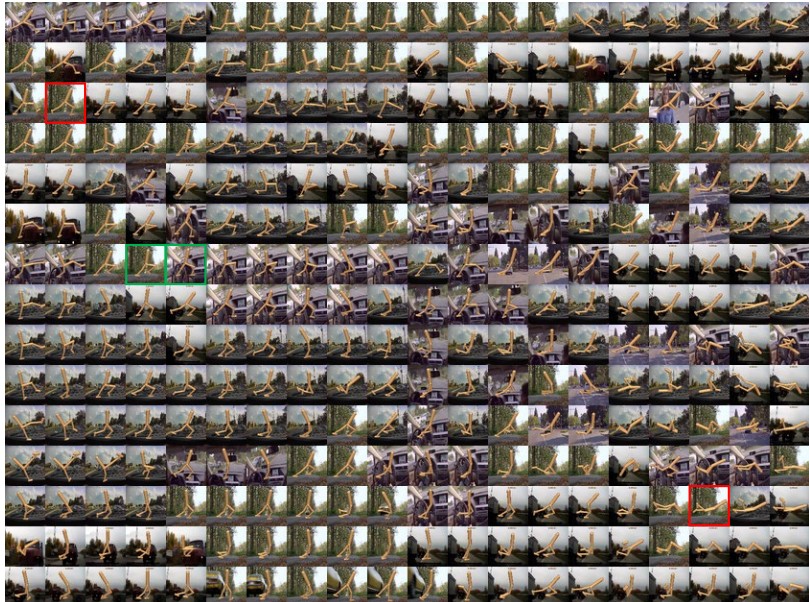

Figure 7: We visualize the representations learned by our method on the Walker Walk task with natural background using t-SNE on a $20 \times 15$ grid. The two images marked in red squares that have different robot configurations and similar backgrounds are far away from each other, while the image pair marked by green squares that have similar robot configurations but different backgrounds, are close to each other.

2015). Fig. 7 shows the visualization of the representations learned by ROTOC on the *Natural Walker Walk* task. Visualizations for the deterministic embeddings and the baselines are shown in Appendix F. The latent space of the learned compressed representation organizes the variation in robot configurations even with natural backgrounds. Specifically, images with similar robot configurations and clearly different backgrounds appear close to each other (e.g. the image pair marked in green), while observations with different robot configurations but similar backgrounds are far away from each other (e.g. the two images marked in red), indicating that the learned representations meaningfully capture task-relevant information related to robot configurations, while filtering out task-irrelevant perturbations.

## 6 Conclusion and Discussion

We presented an information-theoretic representation learning method based on rollout total correlation for image-based RL, ROTOC. By preserving the overall interdependencies among a sequence of observations and actions, ROTOC can learn temporally consistent representations that focus on task-relevant information, increasing the robustness to noise. For achieving a tractable optimization, we derived a lower-bound on the rollout total correlation, which is given by the sum of per-step mutual information. Pursuing our original objective of maximizing rollout total correlation, we proposed chunk-wise mini-batching and adding a more explicit lower bound based on a generative model on top of the discriminative InfoNCE bound. Our experiments show that ROTOC indeed outperforms leading methods in the challenging setting of RL from noisy images. More importantly, our experiments suggest a connection between chunk-based mini-batching, long-term predictability and downstream performance, supporting our hypotheses that chunk-wise mini-batching can increase total correlation, and that maximizing total correlation can result in better representations.

However, by approximating rollout total correlation by a sum of per-step mutual information, our lower bounds may lose some incentive to capture long-term correlations. While the per-step objectives are easier to integrate in existing RL frameworks and the learned transition model could be used for model-based RL in future work, it would also be interesting to explore objectives that better capture multi-step correlations based on multi-step prediction models.

## Acknowledgments

This work was supported by the National Key Research and Development Program of China - Project number 2024YFB4708000, the German Research Foundation (DFG) - Project number PE 2315/18-1, and the German Federal Ministry of Research, Technology and Space (BMFTR) - Project number 01IS23057B. This project has been supported by a hardware donation by NVIDIA through the Academic Grant Program.

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

# A  Bound Derivation

## A.1  Derivation on Eq. 4

In this section, we show how to obtain the lower bound in Eq. 4. We first introduce the entropy $H(\boldsymbol{a}_{1:T-1})$ to the definition of the total correlation in Eq. 2,

$$
\begin{aligned}
\mathcal{C}(\boldsymbol{z}_1; \boldsymbol{a}_1; \cdots; \boldsymbol{a}_{T-1}; \boldsymbol{z}_T) &= \mathbb{E}_{p(\boldsymbol{z}_{1:T}, \boldsymbol{a}_{1:T-1})} \left[ \log \frac{p(\boldsymbol{z}_{1:T}, \boldsymbol{a}_{1:T-1})}{\prod_{t=1}^T p(\boldsymbol{z}_t) \prod_{t=1}^{T-1} p(\boldsymbol{a}_t)} \right] + H(\boldsymbol{a}_{1:T-1}) - H(\boldsymbol{a}_{1:T-1}) \\
&= \mathbb{E}_{p(\boldsymbol{z}_{1:T}, \boldsymbol{a}_{1:T-1})} \left[ \log \frac{p(\boldsymbol{z}_{1:T}, \boldsymbol{a}_{1:T-1})}{p(\boldsymbol{a}_{1:T-1}) \prod_{t=1}^T p(\boldsymbol{z}_t)} \right] + \mathbb{D}_{\mathrm{KL}} \left( p(\boldsymbol{a}_{1:T-1}) \, \| \, \prod_{t=1}^{T-1} p(\boldsymbol{a}_t) \right) \quad (9) \\
&\geq \mathbb{E}_{p(\boldsymbol{z}_{1:T}, \boldsymbol{a}_{1:T-1})} \left[ \log \frac{p(\boldsymbol{z}_{1:T} | \boldsymbol{a}_{1:T-1})}{\prod_{t=1}^T p(\boldsymbol{z}_t)} \right]
\end{aligned}
$$

where the inequality follows from the non-negativity of the KL divergence. We then introduce a variational distribution

$$
q(\boldsymbol{z}_{1:T} | \boldsymbol{a}_{1:T-1}) = p(\boldsymbol{z}_1) \prod_{t=1}^{T-1} p(\boldsymbol{z}_{t+1} | \boldsymbol{z}_t, \boldsymbol{a}_t), \tag{10}
$$

which corresponds to a Markovian approximation of the conditional distribution $p(\boldsymbol{z}_{1:T} | \boldsymbol{a}_{1:T-1})$. We introduce Eq. 10 into Eq. 9 to obtain step-wise losses,

$$
\begin{aligned}
\mathcal{C}(\boldsymbol{z}_1; \boldsymbol{a}_1; \cdots; \boldsymbol{a}_{T-1}; \boldsymbol{z}_T) &\geq \mathbb{E}_{p(\boldsymbol{z}_{1:T}, \boldsymbol{a}_{1:T-1})} \left[ \log \frac{q(\boldsymbol{z}_{1:T} | \boldsymbol{a}_{1:T-1})}{\prod_{t=1}^T p(\boldsymbol{z}_t)} \right] \\
&= \mathbb{E}_{p(\boldsymbol{z}_{1:T}, \boldsymbol{a}_{1:T-1})} \left[ \log \frac{p(\boldsymbol{z}_1) \prod_{t=1}^{T-1} p(\boldsymbol{z}_{t+1} | \boldsymbol{z}_t, \boldsymbol{a}_t)}{p(\boldsymbol{z}_1) \prod_{t=1}^{T-1} p(\boldsymbol{z}_{t+1})} \right] = \sum_{t=1}^{T-1} I(\boldsymbol{z}_{t+1}; \boldsymbol{z}_t, \boldsymbol{a}_t).
\end{aligned} \tag{11}
$$

## A.2  Derivation on Eq. 5

For obtaining a lower bound of the per-step mutual information, we start by replacing the entropy $H(\boldsymbol{z}_{t+1})$ with the conditional entropy $H(\boldsymbol{z}_{t+1} | \boldsymbol{c}_{t+1})$,

$$
\begin{aligned}
I(\boldsymbol{z}_{t+1}; \boldsymbol{z}_t, \boldsymbol{a}_t) &= H(\boldsymbol{z}_{t+1}) - H(\boldsymbol{z}_{t+1} | \boldsymbol{z}_t, \boldsymbol{a}_t) \geq H(\boldsymbol{z}_{t+1} | \boldsymbol{c}_{t+1}) - H(\boldsymbol{z}_{t+1} | \boldsymbol{z}_t, \boldsymbol{a}_t) \\
&= \mathbb{E}_{p(\boldsymbol{z}_{t+1}, \boldsymbol{c}_{t+1}, \boldsymbol{z}_t, \boldsymbol{a}_t)} \left[ \log \frac{p(\boldsymbol{z}_{t+1} | \boldsymbol{z}_t, \boldsymbol{a}_t)}{p(\boldsymbol{z}_{t+1} | \boldsymbol{c}_{t+1})} \right],
\end{aligned} \tag{12}
$$

where the inequality holds because additional information can not increase entropy, $H(\boldsymbol{z}_{t+1}) \geq H(\boldsymbol{z}_{t+1} | \boldsymbol{c}_{t+1})$.

Then, we use a variational distribution $q(\boldsymbol{z}_{t+1} | \boldsymbol{z}_t, \boldsymbol{a}_t)$ to approximate the conditional distribution $p(\boldsymbol{z}_{t+1} | \boldsymbol{z}_t, \boldsymbol{a}_t)$, and then we obtain a KL-based lower bound in Eq. 5:

$$
\begin{aligned}
I(\boldsymbol{z}_{t+1}; \boldsymbol{z}_t, \boldsymbol{a}_t) &\geq \mathbb{E}_{p(\boldsymbol{z}_{t+1}, \boldsymbol{c}_{t+1}, \boldsymbol{z}_t, \boldsymbol{a}_t)} \left[ \log \frac{q(\boldsymbol{z}_{t+1} | \boldsymbol{z}_t, \boldsymbol{a}_t)}{p(\boldsymbol{z}_{t+1} | \boldsymbol{c}_{t+1})} \right] + \mathbb{D}_{\mathrm{KL}} \left( p(\boldsymbol{z}_{t+1} | \boldsymbol{z}_t, \boldsymbol{a}_t) \, \| \, q(\boldsymbol{z}_{t+1} | \boldsymbol{z}_t, \boldsymbol{a}_t) \right) \\
&\geq -\mathbb{E}_{p(\boldsymbol{z}_{t+1}, \boldsymbol{c}_{t+1}, \boldsymbol{z}_t, \boldsymbol{a}_t)} \left[ \log \frac{p(\boldsymbol{z}_{t+1} | \boldsymbol{c}_{t+1})}{q(\boldsymbol{z}_{t+1} | \boldsymbol{z}_t, \boldsymbol{a}_t)} \right] = -\mathbb{E}_p \left[ \mathbb{D}_{\mathrm{KL}} \left( p(\boldsymbol{z}_{t+1} | \boldsymbol{c}_{t+1}) \, \| \, q(\boldsymbol{z}_{t+1} | \boldsymbol{z}_t, \boldsymbol{a}_t) \right) \right].
\end{aligned} \tag{13}
$$

# B  Implementation Details

In this Section, we explain the implementation details for ROTOC in the Deepmind Control suite setting.

### B.1 Pixels Preprocessing

Following (Yarats et al., 2021b), we obtain an individual state by stacking 3 consecutive frames, where each frame is an RGB rendering image with size $84 \times 84 \times 3$ from the 0th camera. The pixel values range from $[0, 255]$ and we divide each pixel value by 255 to scale it down to $[0, 1)$ range. We follow Proto-RL (Yarats et al., 2021a) by performing data augmentation by randomly shifting the image by $[-4, 4]$, before we feed states into the encoders.

### B.2 Network Architecture

The network architecture for ROTOC consists of five modules: deterministic encoder, stochastic encoder, transition model, projection head and prediction head. All five modules are built based on common practice (Yarats et al., 2021a; You et al., 2022; Eysenbach et al., 2021) without specific hyperparameter tuning. We also did not tune the network architectures for the baseline methods, but either used the architectures from the original implementation or the same architecture that we used for testing ROTOC. Deviating from the original network architectures was in some cases necessary, particularly when the baseline method was not evaluated for RL from images. In the following, we list and motivate our choice of network architectures and also indicate with which baseline methods they are shared.

- Deterministic encoder (ROTOC, Proto-RL, DB, RPC). We employ the convolutional encoder architecture from Proto-RL (Yarats et al., 2021a) to parametrize both the online and target deterministic encoders, $\phi_o$ and $\phi_m$. Each deterministic encoder consists of four convolutional layers following a single fully connected layer with Conv(filter=32, kernel-size=3, stride=2) $\rightarrow$ Conv(filter=32, kernel-size=3, stride=1) $\rightarrow$ Conv(filter=32, kernel-size=3, stride=1) $\rightarrow$ Conv(filter=32, kernel-size=3, stride=1) $\rightarrow$ FCN(units=50) architecture. The ReLU activation function is used after each convolutional layer.

- Stochastic encoder (ROTOC, RPC, DB). The architecture of the stochastic encoder is based on the state encoder from RPC (Eysenbach et al., 2021) with a modified hidden dimension. We change the hidden dimension from 256 to larger 1024 since ROTOC focuses on extracting embeddings from high-dimensional images rather than low-dimensional proprioception states in RPC. Specifically, the online stochastic encoder $g_o$ is parameterized by a 2-layer fully-connected network with FCN(units=1024) $\rightarrow$ FCN(units=100) architecture and ReLU hidden activations. Its output is divided into the mean $\boldsymbol{\mu}_o \in \mathbb{R}^{50}$ and the standard deviation $\boldsymbol{\sigma}_o \in \mathbb{R}^{50}$ of the diagonal Gaussian distribution $g_o(\boldsymbol{z}_t|\boldsymbol{c}_t)$. The target stochastic encoder $g_m$ shares the same network architecture as the online stochastic encoder.

- Transition model (ROTOC, RPC, DB). We employ the architecture of the transition model from CoDy (You et al., 2022) with one minor difference to parametrize the transition model. Namely, to match the output dimension of the stochastic encoder, we change the output dimension from 50 to 100 for outputting the mean and standard deviation vectors of a diagonal Gaussian distribution. Specifically, the transition model $q_\psi$ consists of three fully connected layers with FCN(units=1024) $\rightarrow$ FCN(units=1024) $\rightarrow$ FCN(units=100) architecture and ReLU hidden activations. Its output is divided into the mean $\boldsymbol{\mu}_\psi \in \mathbb{R}^{50}$ and standard deviation $\boldsymbol{\sigma}_\psi \in \mathbb{R}^{50}$ of the diagonal Gaussian distribution $q_\psi(\boldsymbol{z}_{t+1}|\boldsymbol{z}_t, \boldsymbol{a}_t)$.

- Projection head (ROTOC, Proto-RL). The projection head $d_v$ is just a single linear layer which maps the concatenated bottleneck variable $\boldsymbol{z}_t \in \mathbb{R}^{50}$ and action $\boldsymbol{a}_t \in \mathbb{R}^{|\mathcal{A}|}$ into a 50-dimensional vector. The projection head is based on the projector from Proto-RL, but accounts for the different input dimensions.

- Prediction head(ROTOC, Proto-RL). The architecture of prediction heads is based on the predictor from Proto-RL (Yarats et al., 2021a) with modified hidden dimension. The online and target prediction heads, $f_o$ and $f_m$, both consist of two fully-connected layers with FCN(units=1024) $\rightarrow$ FCN(units=50) architecture and ReLU hidden activations.

### B.3 SAC Architecture (ROTOC, RPC, ...)

We employ the publicly released standard Pytorch implementation (Yarats et al., 2021b) of SAC. We use all hyperparameters of SAC from (Yarats et al., 2021b), except for the replay buffer capacity, which we set to a smaller $10^5$ by following common practice in image-based RL (Yarats et al., 2021a; You et al., 2022). A smaller buffer can decrease memory costs for storing image observations. All hyperparameters of SAC are fixed across tasks and shown in Table B.1. We refer to (Yarats et al., 2021b) for more detailed descriptions of the implementation of SAC.

Table B.1: Shared hyperparameters across tasks

| Parameter | Value |
|---|---|
| Replay buffer capacity | 100 000 |
| Optimizer | Adam |
| Critic Learning rate | $10^{-3}$ |
| Critic Q-function EMA | 0.01 |
| Critic target update freq | 2 |
| Actor learning rate | $10^{-3}$ |
| Actor update frequency | 2 |
| Actor log stddev bounds | [-10 2] |
| Temperature learning rate | $10^{-3}$ |
| Initial steps | 1000 |
| Discount | 0.99 |
| Initial temperature | 0.1 |
| Learning rate for $\phi_o$, $g_o$, $q_\psi$, $d_v$ and $f_o$ | $10^{-4}$ |
| Encoder and projection model EMA $\tau$ | 0.05 |
| Coefficient $\alpha$ | 0.1 |
| Coefficient $\lambda$ | 0.001 |
| Chunk length | 2 |

### B.4 The Deepmind Control Suite Setting

We evaluate our method ROTOC on a set of challenging continuous control tasks from the commonly used benchmark Deepmind Control Suite. We set the episode length of each task to 1000 steps. Following PlaNet (Hafner et al., 2019b), we treat the number of action repeats as a hyperparameter of the agent. We use the number of action repeats from PlaNet (Hafner et al., 2019b) for each task, which is shown in Table B.2.

Table B.2: Task-specific hyperparameters

| Task | ActionRepeats | Batchsize |
|---|---|---|
| Ball-in-cup Catch | 4 | 256 |
| Cartpole Swingup-sparse | 8 | 256 |
| Reacher Easy | 4 | 256 |
| Cartpole Swingup | 8 | 256 |
| Walker Walk | 2 | 128 |
| Cheetah Run | 4 | 256 |

### B.5 Other Hyperparameters

Following Proto-RL, the learning rate for optimizing networks $\phi_o$, $g_o$, $q_\psi$, $d_v$ and $f_o$ is set to $10^{-4}$, and the coefficient of the exponential moving average used for updating target networks $\phi_m$, $g_m$ and $f_m$ is set to 0.05. Following CoDy (You et al., 2022), we treat batch size as a hyperparameter to the agent. We tested

ROTOC with two batch sizes, 128 and 256, for each task. Based on empirical results, we use a batch size of 256 for all tasks, except for the Walker Walk task, which uses a batch size of 128 (see Table A.2).

There are three specific hyperparameters of ROTOC which are chosen by performing hyperparameter tuning on standard Cartpole Swingup task and subsequently fixed for all tasks, namely coefficient $\alpha$ and $\lambda$ as well as the chunk length $L$. We provide a comprehensive overview of all hyperparameters of ROTOC in Table B.1 and Table B.2.

### B.6 InfoNCE Bound Implementation

In practice, we randomly sample a minibatch of (sequential) transitions $(s_t, a_t, s_{t+1})$, and obtain a minibatch of positive samples $(z_t, a_t, z_{t+1})$ by encoding the minibatch using our models. For a given positive sample, the negative sample set $z_{t+1}^*$ is constructed by using all other embeddings $z_{t+1}$ of the same mini-batch.

### B.7 Baseline Implementation

- **Proto-RL** We obtain the results for Proto-RL by performing the original implementation provided by (Yarats et al., 2021a) with one difference. We do not allow task-agnostic pre-training of Proto-RL to facilitate fair comparison to our setup. Instead, we perform the gradient updates of the representations and SAC agent from the first step of training by jointly optimizing Proto-RL objectives ($L_{SSL}$ and $L_{RL}$) given the task information.

- **DB** We adapt DB to the off-policy continuous control setting. Similar to ROTOC, we employ the convolutional encoder architecture from Proto-RL (Yarats et al., 2021a) to parametrize the observation encoder for DB. The representation posterior and the prediction head of DB are based on the transition model from CoDy (You et al., 2022) with modified output dimension 100. To achieve as good performance as possible for DB, we perform hyperparameter tuning to select suitable $\alpha_2$ and $\alpha_3$ in the information bottleneck objective $L_{DB}$ of DB and an additional hyperparameter $\beta$ which augments DB bonus with environment rewards.

- **RPC** As RPC do not provide an official implementation for image-based control tasks, we implement it by ourselves by following (Eysenbach et al., 2021) as close as possible. Similar to ROTOC, we also employ the convolutional encoder architecture from Proto-RL (Yarats et al., 2021a) following the original stochastic state encoder architecture from RPC to parametrize pixel observation encoder for RPC. The dynamic model of RPC is based on the transition model from CoDy (You et al., 2022) with modified output dimension 100. We select the KL constraint for RPC by hyperparameter tuning to achieve good performance. Specifically, we define a set of the KL constraint values, $[1.0, 3.0, 5.0]$, and perform a grid search over it on the Cartpole Swingup task, and find the optimal KL constraint for RPC is 3.0.

- **Plan2explore** We use the official implementation provided by (Pathak et al., 2017) to acquire the results for Plan2explore.

To ensure a fair comparison, we use the above mentioned implementation of SAC for all methods in our experiments, except for Plan2Explore which follows its original implementation and learns a policy based on a model-based RL method Dreamer (Hafner et al., 2019a). The performance of each algorithm is evaluated by computing an average return over 10 episodes every 10K environment steps. For each method, we perform one gradient update per environment step for all tasks to ensure a fair comparison. Notably, we use the fixed chunk length $L = 2$ for our method across all tasks and the number of samples we used per gradient update step is always twice the batch size. We set the batch size to 512 for Proto-RL by following the official implementation (Yarats et al., 2021a). We set the batch size to 512 for DB and RPC by following Proto-RL and CURL. For Plan2explore, we use the amount of data per gradient step from its official implementations. Our method uses the same amount of samples per gradient step as all model-free baselines on the majority of tasks, except for the Walker Walk task where we use a smaller number of samples.

### B.8 Robustness to Noisy Observations

For this experiment, we add pixel-level Gaussian white noise to the background of each rendered image. The main body of the manipulated object in each task remains undisturbed for efficient observation.

### B.9 Robustness to Natural Background

Following (Zhang et al., 2020), the background of the Mujoco tasks are replaced with by videos randomly sampled from the driving car class in the Kinetics dataset. The backgrounds of the Mujoco tasks are constantly changing during training and evaluation.

### B.10 Algorithm

The training procedure of MTC is presented in Algorithm 1. The algorithm alternates between collecting new experiences from the environment, and updating the parameters of the ROTOC model, the actor and critic networks of SAC.

---

**Algorithm 1:** Training Algorithm for ROTOC

---

**Initialize:** The ROTOC model, actor and critic networks, replay buffer $\mathcal{D}$, Batch size $B$, Chunk length $L$

**for** *each training step* **do**

    collect experience $(\boldsymbol{s}_t, \boldsymbol{a}_t, r_t, \boldsymbol{s}_{t+1})$ and add it to the replay buffer

    **for** *each gradient step* **do**

        Sample a minibatch of sequence chunks $\{(\boldsymbol{s}_t^i, \boldsymbol{a}_t^i, r_t^i, \boldsymbol{s}_{t+1}^i)_{t=k}^{L+k-1}\}_{i=1}^B \sim \mathcal{D}$ from replay buffer.

        Generate $\{(\boldsymbol{c}_t^i, \boldsymbol{c}_{t+1}^i, \boldsymbol{z}_t^i, \boldsymbol{z}_{t+1}^i)_{t=k}^{L+k-1}\}_{i=1}^B$ following Eq.3.

        Compute augmented reward $r_{\mathrm{aug}}(\boldsymbol{s}_t, \boldsymbol{a}_t)$ by following Eq. 8.

        Update the online deterministic and stochastic encoders, the latent transition model, the projection head, the online prediction model and the weight transformation matrix by following Eq. 7.

        Update the actor and critic networks of SAC.

        Update the target deterministic and stochastic encoders and the target prediction head.

    **end**

**end**

---

## C Additional Performance and Robustness Evaluation

Fig. C.1 and Fig. C.2 show the performance of our method and baselines on noisy and natural Mujoco tasks, respectively. Fig. C.3 shows the performance of our method and baselines on all six Mujoco tasks in all three settings to see the performance drops caused by the distractive information.

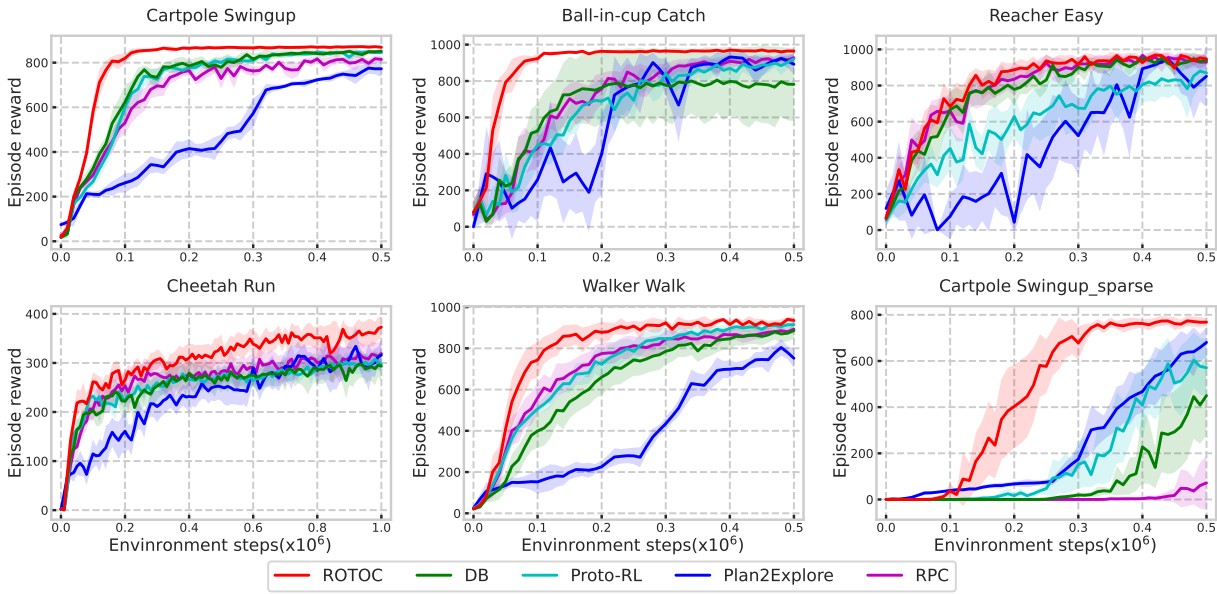

Figure C.1: Robustness comparisons on six noisy Mujoco tasks. Our method performs best on all tasks compared to the baselines.

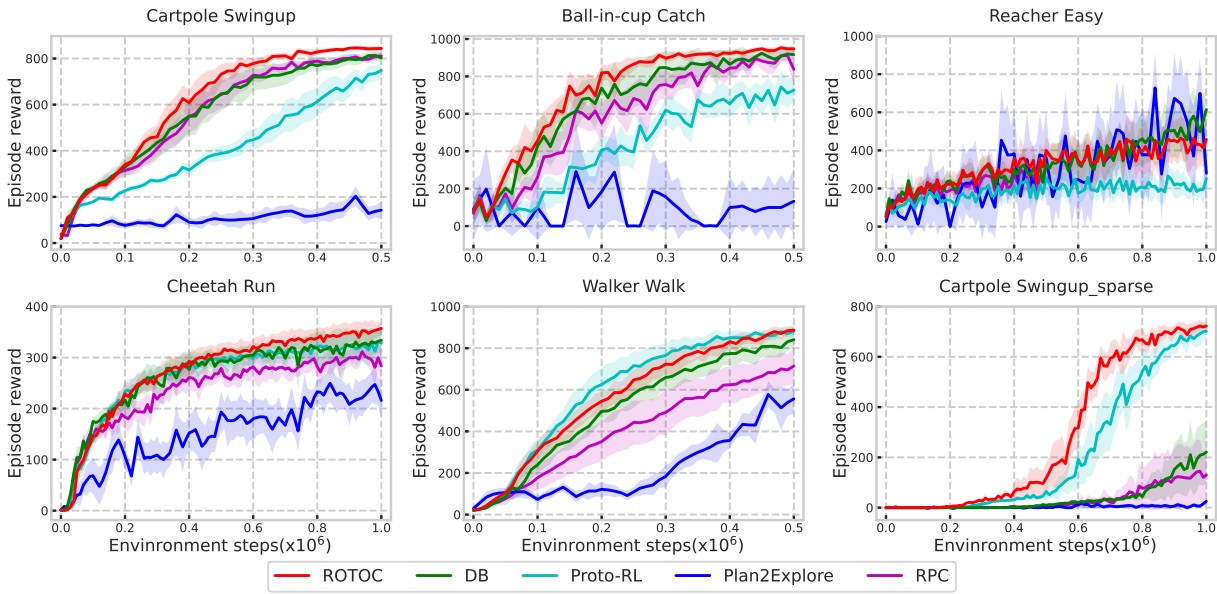

Figure C.2: Performance comparisons on six natural Mujoco tasks. Our method outperforms other methods in robustness to natural video backgrounds.

We also compare the performance achieved by our method and baselines at 500K environment steps on the noisy or natural tasks with the final performance evaluated on the standard tasks for directly checking the

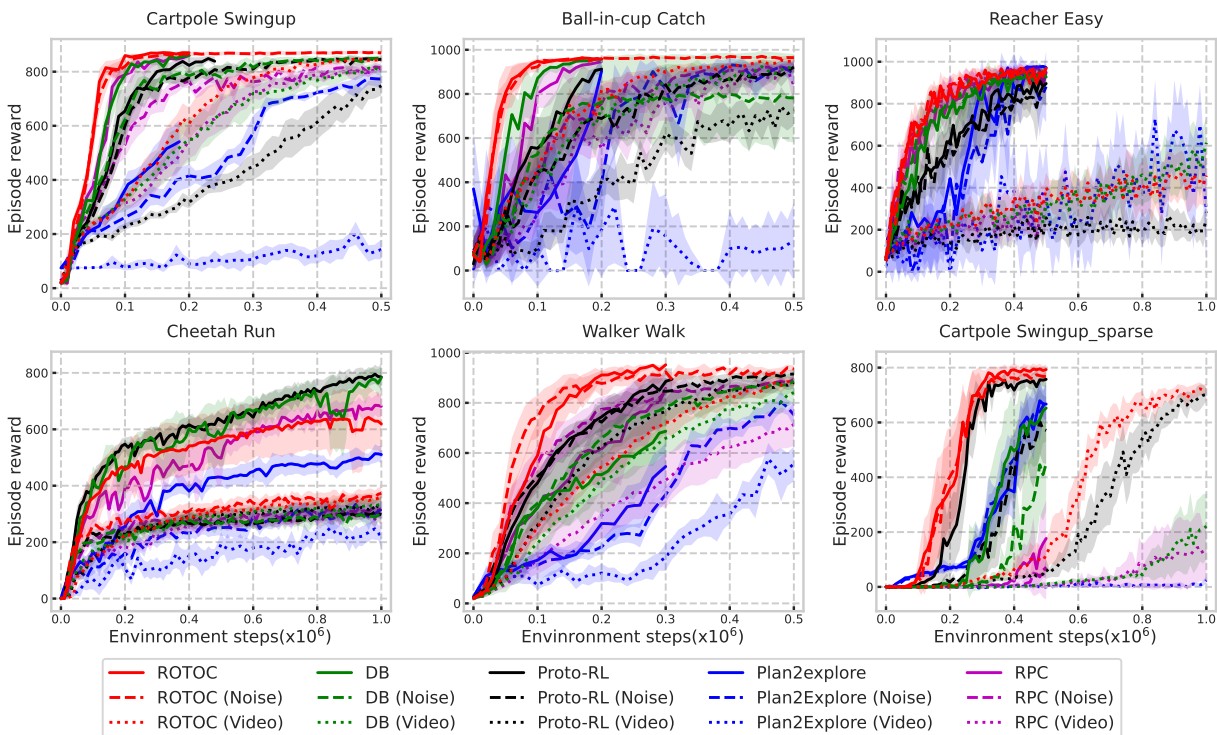

Figure C.3: We show the performance of our method and baselines on all six Mujoco tasks in all three settings to directly compare the performance drops caused by the presence of distractive information ( both white noise and natural video backgrounds).

Table C.1: Scores achieved by our method (mean and standard error for 10 seeds) and baselines at 500k environment steps on noisy Mujoco tasks and natural Mujoco tasks, with the bracketed scores achieved by the corresponding methods at 500K environment steps or the last values we evaluated on standard Mujoco tasks. The dagger (†) indicates that the scores in parentheses are achieved at less than 500K environmental steps. The bold font indicates that the lower-bound on the reward (based on the given standard error intervals) for the given method, is larger or equal than the respective upper bound for every other exploration method.

| 500K step scores | | ROTOC(Ours) | DB | Proto-RL | Plan2Explore | RPC |
|---|---|---|---|---|---|---|
| Noisy Mujoco tasks | Cartpole Swingup | $\mathbf{869\pm 6}$ $(870 \pm 3^{\dagger})$ | $851 \pm 7$ $(860 \pm 5^{\dagger})$ | $846\pm 5$ $(823 \pm 24^{\dagger})$ | $773\pm28$ $(541 \pm 63^{\dagger})$ | $814\pm13$ $(861 \pm 6^{\dagger})$ |
| | Ball-in-cup Catch | $\mathbf{964 \pm 7}$ $(957 \pm 3^{\dagger})$ | $782 \pm 155$ $(957 \pm 5^{\dagger})$ | $922\pm 16$ $(913 \pm 26^{\dagger})$ | $892\pm43$ $(904 \pm 50^{\dagger})$ | $928\pm20$ $(947 \pm 8^{\dagger})$ |
| | Reacher Easy | $\mathbf{947\pm 20}$ $(969 \pm 14)$ | $\mathbf{939 \pm 19}$ $(927 \pm 36)$ | $\mathbf{869\pm 63}$ $(896 \pm 76)$ | $\mathbf{851\pm 132}$ $(892 \pm 6)$ | $\mathbf{930\pm 29}$ $(949 \pm 20)$ |
| | Cheetah Run | $\mathbf{332\pm 20}$ $(570 \pm 79)$ | $265 \pm 20$ $(577 \pm 91)$ | $268\pm 13$ $(631 \pm 20)$ | $251\pm 22$ $(430 \pm 25)$ | $282\pm18$ $(498 \pm 74)$ |
| | Walker Walk | $\mathbf{935\pm 11}$ $(952 \pm 3^{\dagger})$ | $886 \pm 30$ $(687 \pm 120^{\dagger})$ | $\mathbf{915\pm 15}$ $(885 \pm 41^{\dagger})$ | $752\pm36$ $(546 \pm 72^{\dagger})$ | $893\pm14$ $(858 \pm 41^{\dagger})$ |
| | Cartpole Swingup-sparse | $\mathbf{768\pm 14}$ $(793 \pm 10)$ | $450 \pm 133$ $(653 \pm 94)$ | $571\pm 95$ $(757\pm 23)$ | $680\pm 51$ $(667 \pm 51)$ | $72\pm 91$ $(177 \pm 107)$ |
| Natural Mujoco tasks | Cartpole Swingup | $\mathbf{844\pm 6}$ $(870 \pm 3^{\dagger})$ | $803 \pm 16$ $(860 \pm 5^{\dagger})$ | $748\pm 28$ $(823 \pm 24^{\dagger})$ | $143\pm28$ $(541 \pm 63^{\dagger})$ | $812\pm10$ $(861 \pm 6^{\dagger})$ |
| | Ball-in-cup Catch | $\mathbf{947\pm 12}$ $(957 \pm 3^{\dagger})$ | $916 \pm 17$ $(957 \pm 5^{\dagger})$ | $727\pm 53$ $(913 \pm 26^{\dagger})$ | $133\pm136$ $(904 \pm 50^{\dagger})$ | $838\pm72$ $(947 \pm 8^{\dagger})$ |
| | Reacher Easy | $\mathbf{399\pm 57}$ $(969 \pm 14)$ | $327 \pm 65$ $(927 \pm 36)$ | $257\pm 39$ $(896 \pm 76)$ | $\mathbf{362\pm 192}$ $(892 \pm 6)$ | $327 \pm 73$ $(949 \pm 20)$ |
| | Cheetah Run | $\mathbf{308\pm 16}$ $(570 \pm 79)$ | $\mathbf{294 \pm 16}$ $(577 \pm 91)$ | $\mathbf{296\pm 8}$ $(631 \pm 20)$ | $193\pm 32$ $(430 \pm 25)$ | $\mathbf{279\pm12}$ $(687 \pm 120^{\dagger})$ |
| | Walker Walk | $\mathbf{886\pm 15}$ $(952 \pm 3^{\dagger})$ | $841 \pm 31$ $(687 \pm 120^{\dagger})$ | $882\pm 20$ $(885 \pm 41^{\dagger})$ | $557\pm54$ $(546 \pm 72^{\dagger})$ | $714\pm66$ $(858 \pm 41^{\dagger})$ |
| | Cartpole Swingup-sparse | $\mathbf{118\pm 45}$ $(793 \pm 10)$ | $9 \pm 5$ $(653 \pm 94)$ | $65\pm 27$ $(757\pm 23)$ | $0\pm 0$ $(667 \pm 51)$ | $5\pm 4$ $(177 \pm 107)$ |

performance drops. Notably, we just evaluated sample efficiency in the small data regime on some standard tasks, which was enough for our method to converge, but not enough for some of the baselines. In Table C.1 we indicate when the scores are computed at different numbers of steps to avoid wrong conclusions.

# D    Ablation Studies

## D.1    Investigating Chunk Lengths

To evaluate the future predictability of the learned stochastic representations, we first learn representations on the Cartpole Swingup-sparse task with natural backgrounds, and then fix the representations. While using different chunk lengths, we use the same amount of data for sampling from the replay buffer per gradient step. For the fixed stochastic embeddings, we build a forward prediction model parameterized by a neural network, which takes the fixed embedding of the current state at time step $t$, $z_t$, and outputs the mean and the standard deviation of the diagonal Gaussian distribution $d(\boldsymbol{z}_{t+n}|\boldsymbol{z}_t)$ with prediction step $n$. We set the prediction step to 1, 2, 3, or 5. The prediction model consists of three fully connected layers with FCN(units=512) $\rightarrow$ FCN(units=512) $\rightarrow$ FCN(units=100) architecture and ReLU hidden activations. We train the prediction model by maximum-likelihood estimation. We did the same for evaluating the future predictability of the learned deterministic representations.

Table D.1 compares the future predictability of the learned deterministic representations with different chunk lengths on the Cartpole Swingup-sparse task. Compared to using individual transitions, the learned representations by using larger chunk lengths $L = 2, 3$ achieve better prediction at distant prediction steps $n = 3, 5$, while achieving worse prediction at prediction steps $1, 2$. This indicates that maximizing our objective doesn't improve the one-step correlation between latent deterministic embeddings, but causes an increase in the long-term correlations.

Table D.1: We test the future predictability of the learned deterministic representations by optimizing a multi-step predictive model on a data set of latent representations. We obtain larger log-likelihoods with prediction step $t = 3, 5$ and smaller log-likelihoods with prediction step $t = 1, 2$ when using an encoder that was trained with chunk length $L = 2, 3$, compared to using an encoder with chunk length $L = 1$.

| Prediction Steps | 1 | 2 | 3 | 5 |
|---|---|---|---|---|
| L=1,BS=256 | $19.2 \pm 0.3$ | $14.6 \pm 0.1$ | $7.6 \pm 0.1$ | $3.6 \pm 0.2$ |
| L=1,BS=512 | $\mathbf{22.8 \pm 0.2}$ | $\mathbf{15.1 \pm 0.1}$ | $6.4 \pm 0.1$ | $-2.4 \pm 0.1$ |
| L=2,BS=256 | $17.0 \pm 0.3$ | $14.4 \pm 0.2$ | $6.9 \pm 0.0$ | $3.7 \pm 0.2$ |
| L=3,BS=256 | $16.0 \pm 0.2$ | $14.3 \pm 0.1$ | $\mathbf{8.6 \pm 0.1}$ | $\mathbf{4.6 \pm 0.2}$ |
| L=5,BS=256 | $0.5 \pm 0.3$ | $-4.2 \pm 0.1$ | $-11.6 \pm 0.1$ | $-18.4 \pm 0.4$ |

## D.2    Full Results on Three Reward-sparse Tasks

We show the performance of ROTOC and its three ablations, and vanilla SAC on three reward-sparse tasks in three settings in Fig. D.1. ROTOC achieves better or at least comparable performance to its own ablations across all tasks. Moreover, ROTOC significantly outperforms vanilla SAC across all tasks.

## D.3    Effect of KL-based Bound on Improving Predictability

We also compare the future predictability of the learned stochastic and deterministic representations with or without the KL-based lower bound on the standard Cartpole Swingup-sparse task. The results in Table D.2 show that the learned stochastic and deterministic representations by ROTOC are better suited for future prediction compared to the representations learned by Non-KL ROTOC that lacks the KL-based lower bound, indicating that using the KL-based lower bound improves the correlations between the learned representations.

## D.4    Investigating Intrinsic Rewards

### D.4.1    Intrinsic Rewards on Two Subsets of States

Fig. D.2 shows the initial cart position and pole angle in the Cartpole Swingup-sparse task with a random natural background. A pole angle in the range of $[355°, 5°]$ corresponds to an upright pole position, while an

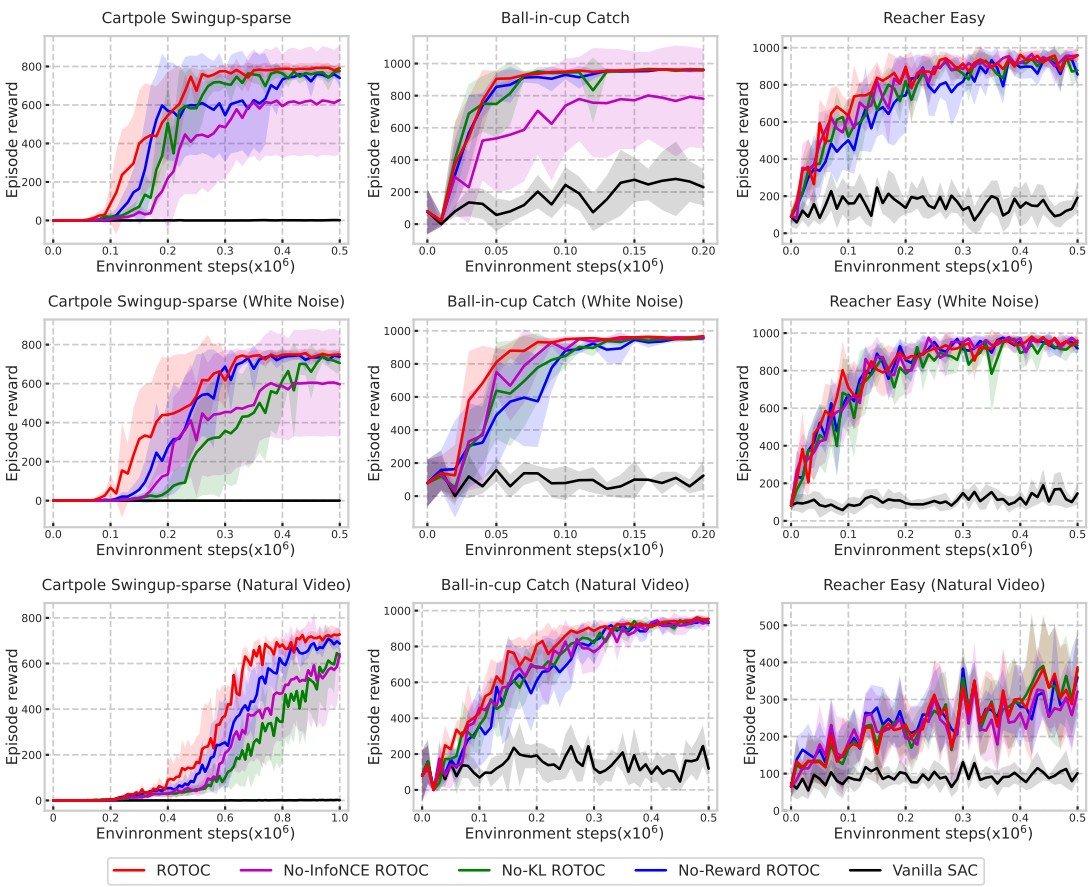

Figure D.1: Performance of ablations on three standard Mujoco tasks (top row), noisy Mujoco tasks (middle row) and natural Mujoco tasks (bottom row). Each subplot shows the average reward for 5 seeds with 90% confidence intervals shading.

Table D.2: The future predictability of the representations learned by ROTOC and Non-KL ROTOC, respectively. The stochastic and deterministic representations learned with the KL bound are better suited for future prediction.

| Prediction Steps | | 1 | 2 | 3 | 5 |
|---|---|---|---|---|---|
| Stochastic Representation | ROTOC | **-80.9 ± 0.4** | **-91.9 ± 0.4** | **-99.6 ± -0.1** | -109.3 ± 0.4 |
| | Non-KL ROTOC | -86.5 ± 0.3 | -94.3± 0.3 | -100.6 ± 0.2 | -109.8± 0.2 |
| Deterministic Representation | ROTOC | **17.0 ± 0.3** | **14.4 ± 0.2** | **6.9± 0.0** | **3.7± 0.2** |
| | Non-KL ROTOC | 9.6 ± 0.1 | 7.4± 0.2 | 3.3 ± 0.3 | 0.3 ± 0.1 |

angle in the range of $[175°, 185°]$ corresponds to a pole pointing downwards. The cart position in the range of $[-0.25, 0.25]$ corresponds to a middle position.

To investigate the effectiveness of our intrinsic reward function, we save the intrinsic reward function every 100,000 steps during training. We then save the replay buffer after training our model from scratch on the task to ensure that the obtained replay buffer is independent of the saved reward function. We sampled a fixed data set from the replay buffer, and split the data set into two, where one of the subsets only contains states with a pole angle in the range of $[355°, 5°]$, and the other subset only contains states with a pole angle in the range of $[175°, 185°]$. For each subset, we sample a minibatch of states and compute the intrinsic rewards of the drawn samples by using the saved reward function.

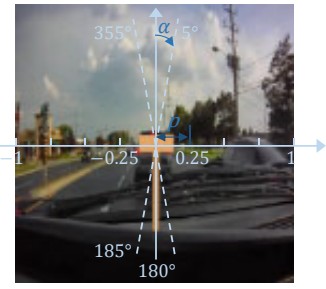

Figure D.2: The pole is initialized pointing downwards with the pole angle $\alpha = 180°$ and located in the middle with the position $p = 0$ in the Cartpole Swingup-sparse task.

### D.4.2 Intrinsic Rewards on Trajectories

We also compute the intrinsic rewards along trajectories sampled from the saved replay buffer. We sample a trajectory and then compute the intrinsic rewards of this trajectory by using the reward functions saved at the different phases of the training. Specifically, we use the reward function saved at 200K environment steps where the policy achieves a reward of around 0 (see Fig. C.2), the reward function saved at 600K environment steps where the reward achieved by the policy is increasing rapidly, and the reward function saved at 1000K environment steps where the policy becomes good at balancing the pole. Fig. D.3, Fig. D.4 and Fig. D.5 show the evolution of the intrinsic reward along an near-optimal trajectory. Regardless of the distractive backgrounds, the reward functions at 200K and 600K environment steps tend to allocate higher intrinsic rewards to images that show an upright pole position, than images that show a pole pointing downwards. The reward function saved at 1000K steps does not show this tendency anymore, since these images that show an upright pendulum become more common in the replay buffer.

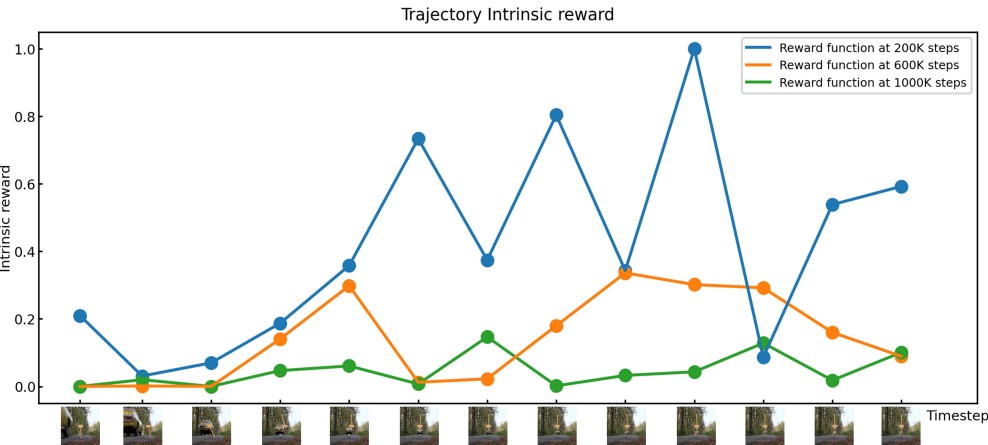

Figure D.3: Evolution of intrinsic rewards along the first of three near-optimal trajectories sampled from the replay buffer for the CartPole Swingup-sparse task. The dots along the x-axis correspond to observations in the sampled trajectory per 80 time steps. Initially (blue curve), higher rewards are assigned to states with an upright pendulum, which are rarely observed. As training progresses (orange and green curves), the intrinsic rewards for such states decrease as they become less novel.

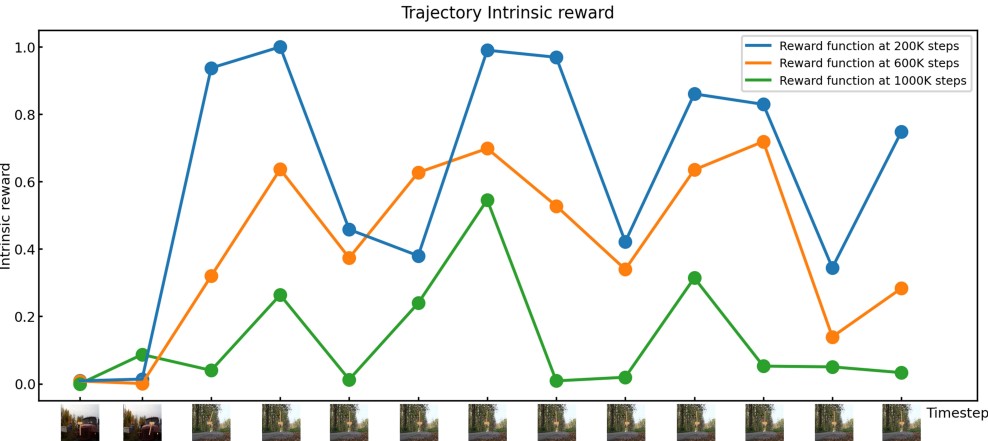

Figure D.4: Evolution of intrinsic rewards along the second of three near-optimal trajectories sampled from the replay buffer for the CartPole Swingup-sparse task. The dots along the x-axis correspond to observations in the sampled trajectory per 80 time steps. Initially (blue curve), higher rewards are assigned to states with an upright pendulum, which are rarely observed. As training progresses (orange and green curves), the intrinsic rewards for such states decrease as they become less novel.

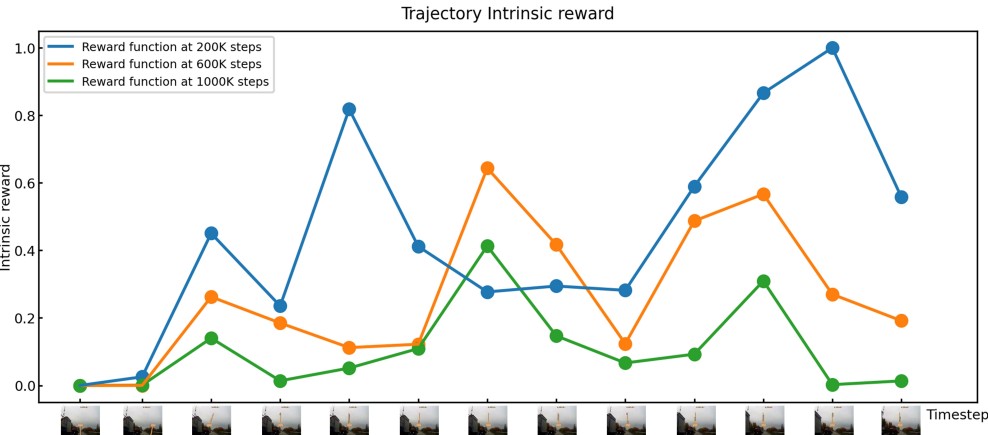

Figure D.5: Evolution of intrinsic rewards along the second of three near-optimal trajectories sampled from the replay buffer for the CartPole Swingup-sparse task. The dots along the x-axis correspond to observations in the sampled trajectory per 80 time steps. Initially (blue curve), higher rewards are assigned to states with an upright pendulum, which are rarely observed. As training progresses (orange and green curves), the intrinsic rewards for such states decrease as they become less novel.

### D.5 Investigating Effect of Chunk-wise sampling on Optimization

In this section, we perform an ablation experiment to investigate whether using chunk-wise mini-batching sampling produces more consistent gradients along trajectories. In our experiment, on the Cartpole Swingup-sparse task, we collected 20 independent fixed trajectories, computed the gradients of the parameters of the encoder with respect to the loss every time step in trajectories, and then calculated the variance of gradients over all time steps during training. We compare the curves of the gradient variance for different chunk lengths in Fig. D.6 and the final gradient variance in Table D.3. While using the same number of data samples, using chunking with $L \geq 2$ achieves a smaller gradient variance than using a chunk length of 1. We also observed that the variance of using individual transitions significantly increases faster than using larger transitions when the number of samples increases. These results demonstrate that using chunk-wise mini-batching sampling makes the gradients along a trajectory more consistent, helping learning globally consistent representations.

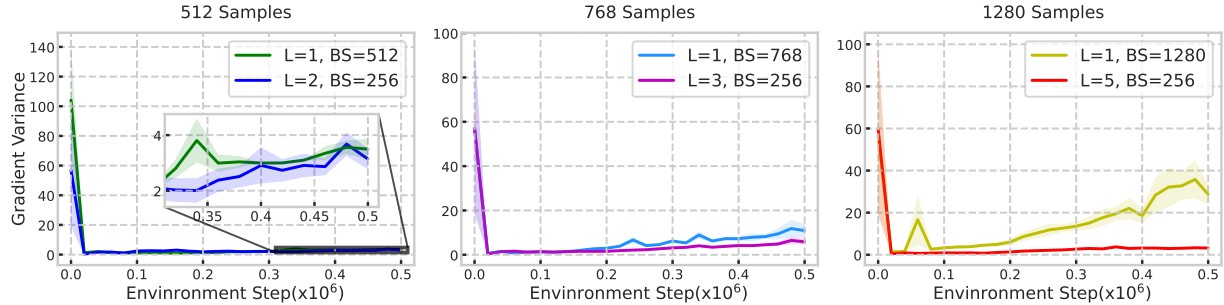

Figure D.6: We compare the variance of the gradients over time steps for different chunk length $L$ and batch size $BS$. The plot shows the mean and standard error for 3 seeds. Using chunking produces a smaller gradient variance along a trajectory than without chunking, when using the same number of samples.

Table D.3: We compare the variance of the gradients over time steps for different chunk length $L$ and batch size $BS$. The table shows the mean and standard error for 3 seeds. When using the same amount of training samples, using sequential transitions achieves smaller gradient variance than using individual transitions ($L = 1$).

| Configurations | 512 samples | | 768 samples | | 1280 samples | |
| --- | --- | --- | --- | --- | --- | --- |
| | L=1, BS=512 | L=2, BS=256 | L=1, BS=768 | L=3, BS=256 | L=1, BS=1280 | L=5, BS=256 |
| Gradient variance | 3.5± 2.0 | **3.1± 1.8** | 10.9± 6.3 | **5.9± 3.3** | 28.7± 16.6 | **3.2± 1.8** |

# E   Model Complexity

We evaluate the model complexity of ROTOC and other methods. Table E.4 shows the number of learnable parameters for all methods. The model complexity of ROTOC is higher than RPC, but is lower than DB and Proto-RL.

Table E.4: Comparison of model complexity. ROTOC has 3.5 million learnable parameters, slightly higher than DB (3.4 million) and lower than DB and Proto-RL.

| Method | ROTOC (Ours) | RPC | DB | Proto-RL |
|---|---|---|---|---|
| Number of Parameters | 3.5M | 3.4M | 4.6M | 5.2M |

# F  Representation Visualization

Fig. F.1, Fig. F.2 and Fig. F.3 visualize the learned representations by RPC, Proto-RL and DB on the natural Walker Walk task, respectively. We also visualize the learned deterministic representation learned by our method on the natural Walker Walk task in Fig F.4. We evaluate learned representations by seeing if representations that have similar robot configurations and clearly different backgrounds appear close to each other. The representations learned by Proto-RL and RPC look worse, but the representations learned by DB and our deterministic embeddings are comparable to the stochastic embedding learned by our method.

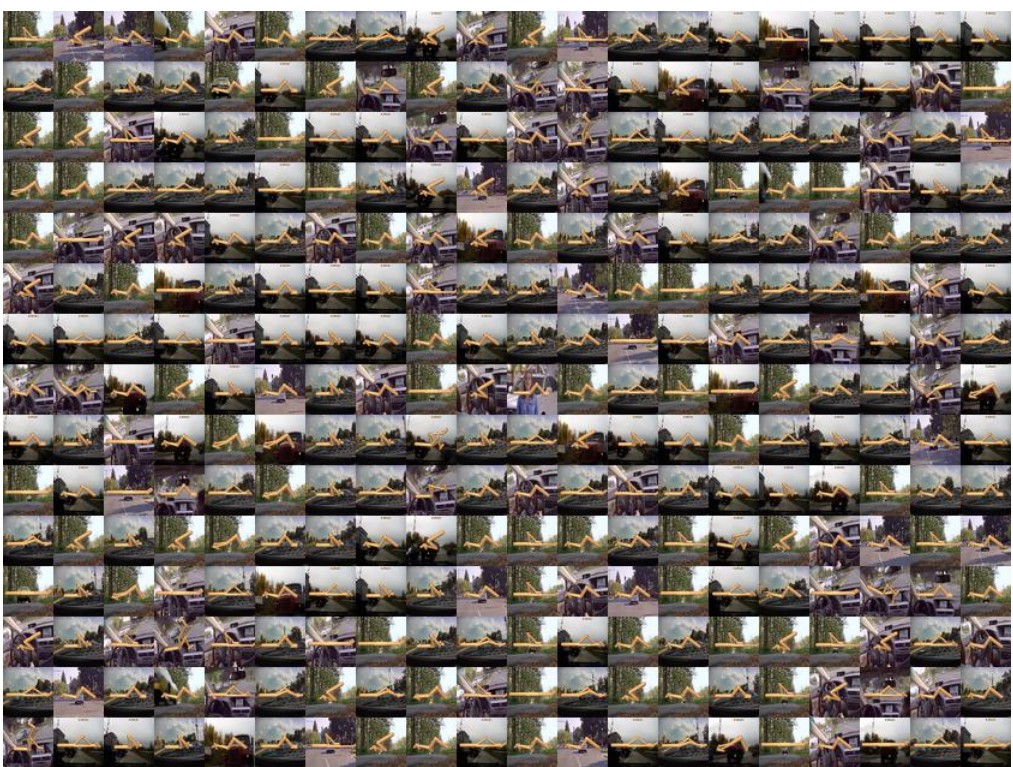

Figure F.1: The representations learned by RPC are visualized using t-SNE on a $20 \times 15$ grid.

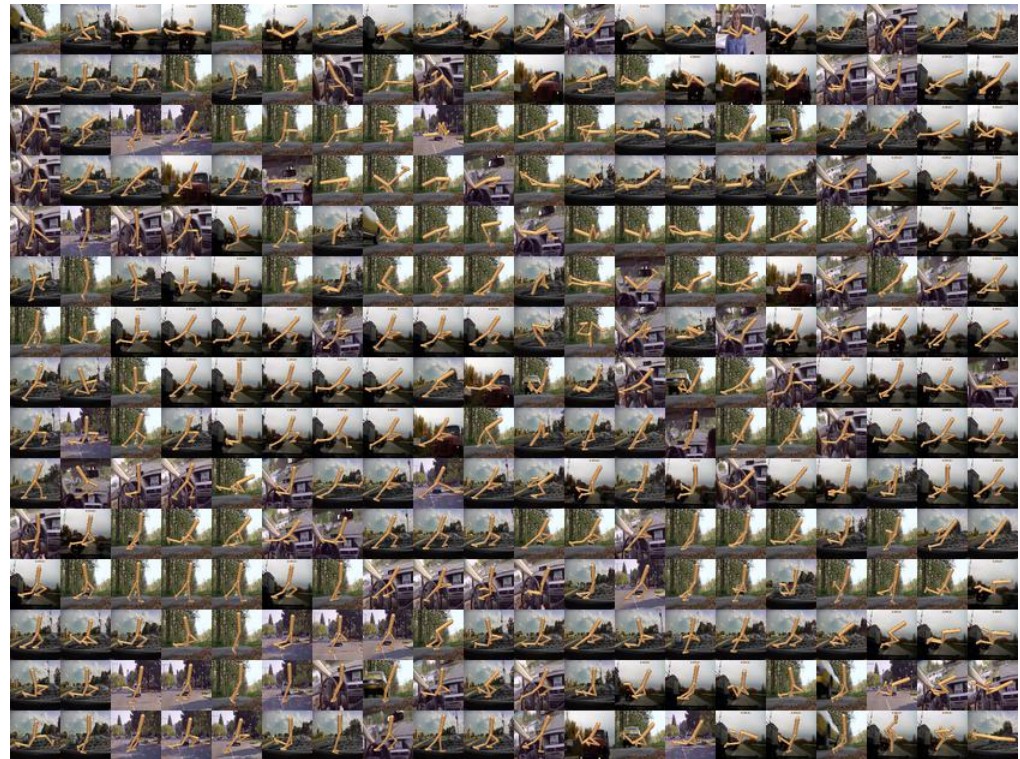

Figure F.2: The representations learned by Proto-RL are visualized using t-SNE on a $20 \times 15$ grid.

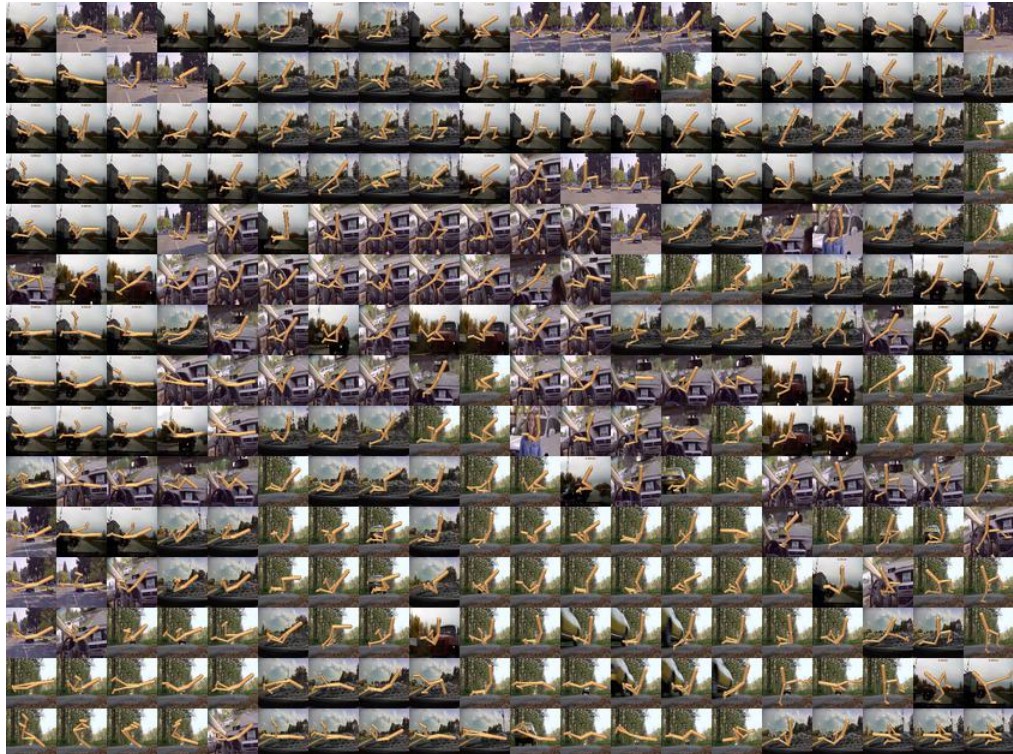

Figure F.3: The representations learned by DB are visualized using t-SNE on a $20 \times 15$ grid.

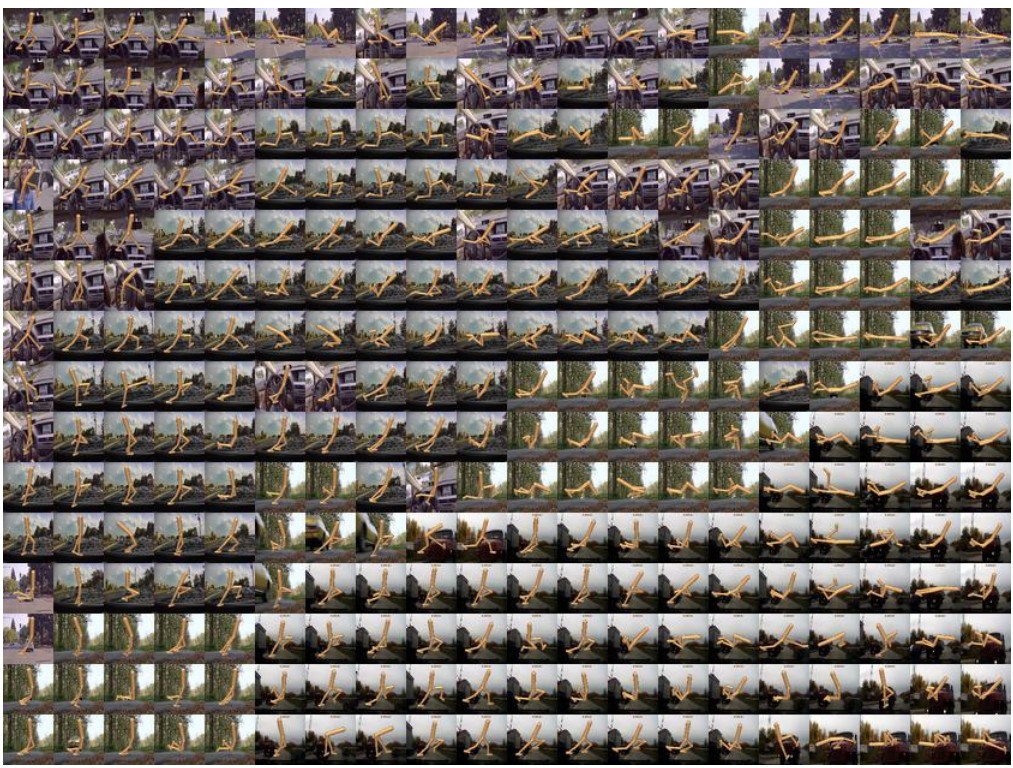

Figure F.4: The deterministic representation learned by our method on the Walker Walk task with natural background are visualized using t-SNE on a $20 \times 15$ grid.

