# OpenReview forum: "Rollout Total Correlation for Deep Reinforcement Learning"
_TMLR — Accepted by TMLR_

### Review · Reviewer_DXqP · 2025-05-12

**Summary Of Contributions:**

This paper proposed a Rollout Total Correlation (ROTOC) as a novel objective for representation learning in reinforcement learning, designed to capture long-term temporal dependencies within trajectories. This paper derive a lower bound by combining two complementary approximations of per-step mutual information: one generative and one discriminative. Additionally, they introduce a chunk-based mini-batching strategy to better preserve multi-step correlations during training.  The Results show that the method significantly improves sample efficiency and robustness to observation noise compared to leading baselines.

**Audience:**

Yes

**Claims And Evidence:**

Yes

**Requested Changes:**

Please refer to above weakness.

**Strengths And Weaknesses:**

>Strengths:
1. The objective is novel
2. The empirical comparison is comprehensive and solid

>Weakness:
1. How close or tight is the proposed lower bound compared to the true rollout total correlation?
2. since earlier states influence future ones. However, the best results are observed at L = 3. What factors limit the performance at higher chunk lengths? Additionally, will different environments have different best perform L?
3. This method involves many NN models. Although the result is promising, how can we fine tune all the models? e.g. determine efficent network size for the models?
4. How is the computaion complexity of the method with such complex models?

---

> ### Author Response · Authors · 2025-06-02
> **Rebuttal by Authors**
>
> Thank you for carefully reviewing our submission, and acknowledging the novelty and sound experiments of our method.
>
> > How close or tight is the proposed lower bound compared to the true rollout total correlation?
>
> Thank you for this question. The tightness of our lower bound depends on the amount of information that we disregard by only considering per-step mutual information, and by the tightness of our lower bound on that mutual information. Regarding the former, we disregard the direct effect of action correlations (how much information can be saved by encoding a pure action trajectory instead of individually encoding the actions per time-step), since we focus on learning state-representations. Furthermore, our variational distribution is a Markovian approximation of an autoregressive model, and thereby only uses the current representation and action ($z_t$ and $a_t$) which prevents us from tightly approximating the actual joint distribution $p(z_{1:T}, a_{1:T})$. Quantifying these effects is difficult, as they can greatly depend on the task, and furthermore the ground-truth total correlation is typically not available.
>     Regarding the second point, our lower bound uses the InfoNCE bound, which is a tight lower bound on the per-step mutual information, when neglecting the effect of limited number of samples and the suboptimality of the discriminator. The actual value in our implementation will still differ from this estimate of the mutual information, since we do not include the constant offset of the InfoNCE bound ($\log B$ with batch size $B$), which does not affect optimization.
>     In summary, we do not expect our lower bound to faithfully capture the total correlation for various reasons. However, for our practical purposes of learning relevant and predictable state representations, our lower bound is a tractable and suitable approximation, as demonstrated in our experiments.
>
> > What factors limit the performance at higher chunk lengths?
>
>  Thank you for this valuable comment. It is not possible to independently evaluate the effect of the chunk length, because we either reduce the variability (number of different trajectories) in the mini-batch, or increase the total number of transitions in the mini-batch. Reducing the number of trajectories in the mini-batch could impair optimization, leading to worse local optima as the gradients are more likely to overfit towards fewer trajectories. If we try to keep the number of trajectories constant, we have to increase the number of individual transitions, which is also expected to affect the performance, and therefore acts as a confounding factor. However, as shown in Table 3, a chunk length of 5 is better than a chunk length of 1, when keeping the total number of transitions constant.
>
> > Will different environments have different best perform L?
>
> Although the chunk length L=3 achieves the best performance on two specific environments, the best chunk length generally varies across environments due to different environmental characteristics. But using an optimal chunk length is not quite important. In our experiments, we obtain good performance even without tuning the chunk length for each task.
>
> > How can we fine-tune all the models?
>
> Thank you for pointing out this problem. Our ROTOC model consists of five modules: deterministic encoder (which is also a part of the policy), stochastic encoder, transition model, projection head, and prediction head. All five modules are built on common practice without specific hyperparameter tuning. We refer to Appendix B.2 for more details on determining the network architecture.
>
> > How is the computation complexity of the method with such complex models?
>
> Thank you for this constructive feedback. In Appendix E, we now evaluate the computational complexity of ROTOC by counting the number of learnable parameters. Our model has around 3.5M parameters, slightly higher than 3.4M of RPC, and lower than 4.6M of DB and 5.2M of Proto-RL. Hence, our algorithm has lower or at least comparable computational complexity than previous approaches. However, we realized that the computational complexity of ROTOC is higher than SAC, since our algorithm uses additional models for learning embeddings.

---

> ### Comment · Action_Editor_7PNL · 2025-06-12
>
> Thank you for all your efforts in reviewing this paper so far.  Have you find the authors' rebuttal satisfactory? Is there any remaining issues? If you need to discuss anything only among reviewers, there is a separate Discussion thread. At the very least, please acknowledge the authors' response..
>
> Thank you,
> Action Editor

---

> > ### Comment · Reviewer_DXqP · 2025-06-20
> > **Thanks for the replay**
> >
> > I thank auther's feedback and address my concerns. I do not have further questions

---

### Review · Reviewer_dHTV · 2025-05-12

**Summary Of Contributions:**

The authors present a method for increasing sample efficiency in reinforcement learning through incorporating temporally consistent representation learning in an off-policy RL setting. They argue that optimizing for total correlation of the state action sequences is beneficial for learning efficient representation of dynamic tasks from image data, propose an encoder architecture and an information theoretic lower bound for such total correlation based on mutual information estimates, incorporate this into a proposed modified reward function and apply SAC with such modified reward function. They demonstrate in image-based robotic tasks how their method improves (in most cases) over baselines.

**Audience:**

Yes

**Claims And Evidence:**

Yes

**Requested Changes:**

1. Can the authors give a more detailed justification for why considering correlation objectives over entire trajectories would help RL policies perform better in robotic tasks (or in any case, learn faster/with less data)? What types of tasks, and for which tasks such objectives are not expected to induce any improvement?
2. Can the authors provide more detail on the specific differences and novelty in their method with respect to other algorithms that already consider mutual information objectives in off-policy RL (e.g. RPC)?
3. Given that RPC also uses SAC as a base implementation, and that it also computes MI estimates between latent state-action pairs and uses them to learn more compact representations, I would expect the empirical performance of RPC and ROTOC to be closer in most tasks. What is the reason for RPC having such poor performance in the experiments chosen when compared to ROTOC? What are the main differences in both implementations that cause these?

**Strengths And Weaknesses:**

## Strengths:
- The contributions and motivation are clearly stated and (relatively) well justified.
- The paper is clear, and the experiments are sufficient to back the conclusions and contribution statements.
- The main idea is intuitively straight-forward.

## Weaknesses:
- I'm not entirely convinced by the motivation (although it is clear, as mentioned). While considering long-term information gain in sequential problems for learning better representations seems reasonable, the reasons for why this should induce an across-the-board improvement of RL algorithms for robotic tasks is not clear to me, and in fact does not seem to be the case from the experimental results (there are clear improvements in some tasks, but it is not obvious to me that these are not due to task selection bias).
- The resulting lower bounds (which make the objective tractable) seem to result in a loss which is essentially reduced to a sum of mutual information between consecutive time-steps. For this reason, the main difference with respect to e.g. RPC (which proposes a mutual information penalty to learn more compact and predictable representations in off-policy RL) seems to be the mini-batch training heuristic, where the data for training is sampled in batches of consecutive trajectory points to bias the learning towards 'longer term' consistency. This seems like a slightly thin contribution given the body of existing work.

---

> ### Author Response · Authors · 2025-06-02
> **Rebuttal by Authors**
>
> Thank you for providing constructive feedback, and acknowledging our work.
>
> > justify why considering long-term correlations helps RL
>
> Thank you for this critical feedback. Although we argue that it is quite intuitive that stable temporally consistent features such as the energy of a closed system, absolute poses of objects, and higher-level indicator features ("object-in-hand'', "gripper-opened'') are often quite powerful for describing the current state and deriving appropriate actions, it is indeed not obvious why exactly they should help in a model-free RL setting. However, even though we don't explicitly rely on predictions during model-free RL, we argue that consistent and predictable representation are still useful as they will result in a smoother mapping from latent representation to optimal action, and may help the agent in exploring more efficiently in noisy environments (even without the auxiliary reward).
>
> > which tasks such objectives are not expected to induce any improvement?
>
> We expect that biasing representation towards long-term consistency may be detrimental for tasks that require highly fluctuating actions, since states with similar representations would need to get mapped to very different actions.
>
> > differences and novelty with respect to other works based on mutual information (e.g. RPC)
>
> There are several significant differences in the motivation and problem setting, the resulting algorithms, as well as the contributions of the different methods.
>
> Motivation and Problem Setting: RPC aims to learn simpler policies by restricting the amount of information that is encoded in the representations, aiming for simpler and more robust behavior. Instead, ROTOC aims to learn more consistent representations for reinforcement learning from images.
>
> The algorithms: RPC uses a common objective to optimize policy and encoder, such that the policy actively aims to reach states where little information needs to be encoded into the latent representation. Instead ROTOC applies a standard SAC actor and critic update, while using the total correlation loss as a separate objective for training the encoder and corresponding models. While our total correlation loss is also derived from a sum of per-step mutual information, it differs from the RPC loss, by using a combination of two lower bounds, based on a generative and discriminative model.
>
> The contributions: RPC introduced sequential mutual information for reinforcement learning and presented a new method based on an upper bound on it. While our KL-based lower bound on the total correlation takes a similar form, we derived it from a different perspective. Furthermore, we combined it with a lower bound based on a discriminative model, introduced and analyzed the simple yet powerful technique of chunk-wise mini-batching, and proposed an auxiliary reward for better exploration based on our total correlation bound.
>
> > the reason for RPC having poor performance
>
> Thank you for this constructive feedback. In our previous implementation of RPC, following RPC, we used a single objective to optimize policy and encoder, and allowed the gradients from the upper bound of the mutual information and the actor to update the parameters of the encoder. We conjecture that passing the gradients from the actor to the encoder degrades performance, since it modifies the encoder when updating the upper bound. In our revision, we now improve the implementation for RPC by stopping the gradient of the actor through the encoder, while allowing the gradients of the critic to backpropagate through the encoder during the critic update. We observe that this updated implementation achieves strong performance (see Figure 4 and Table 1).

---

> > ### Comment · Reviewer_dHTV · 2025-06-16
> > **Acknowledgment**
> >
> > I thank the authors for their reply and the changes made in the paper. I do not have further questions.

---

> ### Comment · Action_Editor_7PNL · 2025-06-12
>
> Thank you for all your efforts in reviewing this paper so far.  Have you find the authors' rebuttal satisfactory? Is there any remaining issues? If you need to discuss anything only among reviewers, there is a separate Discussion thread. At the very least, please acknowledge the authors' response..
>
> Thank you,
> Action Editor

---

### Review · Reviewer_7GyU · 2025-05-24

**Summary Of Contributions:**

The authors study the problem of learning task-relevant representations in image-based reinforcement learning, with a focus on long-term temporal consistency and objectives that incentivize consistency over long horizons.

1.  **ROTOC Objective**:
    * Proposes maximizing the total correlation among latent state embeddings ($z_t$) and actions ($a_t$) within entire trajectories: $\max \mathcal{C}(z_1; a_1; \dots; a_{T-1}; z_T)$.

2.  **Tractable Optimization Framework**:
    * The ROTOC objective is lower-bounded by the sum of per-step mutual information: $\sum_{t=1}^{T-1} I(z_{t+1}; z_t, a_t)$.
    * This per-step $I(z_{t+1}; z_t, a_t)$ is further approximated by combining two complementary lower bounds:
        * A **generative model-based bound** using KL divergence (for predictable and consistent representations) .
        * A **discriminative model-based bound** using InfoNCE loss (avoid collapse and trivial solutions of KL bound): $I_{\omega}(z_{t+1};z_{t},a_{t})$.
    * The final representation learning loss combines these terms:
        $L = \mathbb{E}[\sum_{t=1}^{T-1}\alpha$ $\mathbb{D}\_{KL} ( g_m $(z\_{t+1} $ $|c\_{t+1}) || q\_{\psi} (z\_{t+1}|z_t,a_t))$ $ - $I_{\omega}(z_{t+1};z_t,a_t)]$.

3.  **Chunk-Wise Mini-Batching**:
    * A technique where mini-batches are sampled as sequence chunks $(s_t, a_t, s_{t+1})_{t=k}^{L+k-1}$ of length $L$.
    * This is proposed to better approximate the original ROTOC objective and encourage learning of long-term correlations, despite the per-step decomposition of the bound.
    * Empirically shown to increase long-term predictability and agent performance.

4.  **Intrinsic Reward for Exploration**:
    * An intrinsic reward $r^*(s_t, a_t)$ is derived from the InfoNCE loss term to promote exploration.
    * Specifically, $r^*(s\_t , a\_t ) = $ $ -\mathbb{E}_{p(z\_t, a\_t, z\_{t+1})} $ $[\text{InfoNCE log term} ]$ , with higher values for less predictable transitions.

5.  **New Knowledge from Empirical Validation**:
    * ROTOC achieves improved sample efficiency and final performance on standard image-based Mujoco tasks.
    * Demonstrates enhanced robustness to observational noise (Gaussian white noise and natural video backgrounds) compared to baselines.
    * Ablation studies confirm:
        * Chunk-wise mini-batching (e.g., $L=2,3$) outperforms individual transitions ($L=1$) and leads to better multi-step predictability of representations.
        * Both the KL-based and InfoNCE-based lower bounds, as well as the intrinsic reward, contribute positively to performance. The KL-based bound specifically improves long-term predictability.
        * Learned representations effectively filter out distractors and capture task-relevant features (e.g., robot configurations).

**Audience:**

Yes

**Broader Impact Concerns:**

The research focuses on improving representation learning techniques for reinforcement learning agents, primarily validated in simulated control environments. The specific methodology of maximizing rollout total correlation does not appear to introduce unique ethical concerns beyond those generally applicable to AI and reinforcement learning research. The paper's aim to enhance robustness to visual distractors could be viewed as a positive step towards more reliable AI systems.

**Claims And Evidence:**

Yes

**Requested Changes:**

1.  **Clarification of the Link Between Per-Step Loss and Global ROTOC Objective with Chunk-Wise Mini-Batching**
    * **Specification**: *Would significantly clarify and strengthen the work's claims regarding the optimization of long-term correlations.*
    * **Adjustment**: Elaborate further in Sections 4.3 and 4.5.1 on how optimizing the sum of per-step mutual information objectives, $\sum_{t=1}^{T-1}I(z_{t+1};z_{t},a_{t})$, using chunk-wise mini-batching helps in achieving the global objective of maximizing rollout total correlation $\mathcal{C}(z_{1};a_{1};\cdot\cdot\cdot;a_{T-1};z_{T})$. The current text states chunk-wise mini-batching "may help to converge to local optima with better total correlation" and relies on the intuition that "a stochastic gradient that increases correlation between several subsequent representations [will] favor temporally consistent representations". Expanding on *why* this batching strategy, in conjunction with the per-step loss, is hypothesized to better approximate the optimization of the original long-term objective would be beneficial. For instance, discuss if it encourages smoother representation manifolds over sequences or if the aggregated gradients from chunks better reflect the dependencies across the entire rollout.

2.  **Explicit Details on InfoNCE Negative Sample Strategy**
    * **Specification**: *Would strengthen the work by enhancing reproducibility.*
    * **Adjustment**: In Section 4.4.2 or Appendix B, provide precise details on how the negative samples $z_{t+1}^{*}$ (for the InfoNCE loss $I_{\omega}(z_{t+1};z_{t},a_{t})$) are obtained in practice. The text mentions they are "drawn from the marginal distribution $p(z_{t+1})$", but it would be helpful to clarify if these are, for example, other $z_{t+1}$ samples from the same mini-batch, from a separate memory bank, or another common approximation strategy.

3.  **Brief Acknowledgment of Hyperparameter Tuning Scope for Representation Learning Components**
    * **Specification**: *Would strengthen the work by improving transparency regarding the experimental methodology.*
    * **Adjustment**: Add a sentence in the experimental setup (Section 5.1) or implementation details to explicitly state that the primary representation learning hyperparameters (e.g., coefficient $\alpha$, $\lambda$ for the intrinsic reward, and chunk length $L$) were selected based on performance on a specific task (standard Cartpole Swingup) and then applied consistently across all other tasks and environments.
    * **Reasoning**: This highlights the generalization of the chosen hyperparameter set while implicitly noting that task-specific tuning wasn't exhaustively performed for every environment, providing a complete picture of the tuning strategy.

**Strengths And Weaknesses:**

## Strengths

1.  **Novel and Intuitive Objective**:
    * The core **Rollout Total Correlation (ROTOC)** objective, $\max \mathcal{C}(z_1; a_1; \dots; a_{T-1}; z_T)$, is a well-motivated and principled approach to learn representations that capture long-term temporal consistency. This directly addresses limitations of prior methods focusing on individual transitions.
2.  **Comprehensive and Tractable Method**:
    * The paper successfully translates the intractable ROTOC objective into a practical algorithm by deriving a lower bound based on the sum of per-step mutual information, $\sum_{t=1}^{T-1} I(z_{t+1}; z_t, a_t)$.
    * It employs two complementary lower bounds for the per-step mutual information: a generative (KL-divergence based) model for consistency and a discriminative (InfoNCE based) model for representativeness.
    * The introduction of **chunk-wise mini-batching** is a simple yet effective technique to encourage the learning of longer-term correlations, helping to align the per-step optimization with the global objective.
    * An **intrinsic reward** derived from the InfoNCE loss is integrated to encourage task-specific novelty and exploration.
3.  **Strong Empirical Results and Robustness**:
    * The method demonstrates improved sample efficiency and often superior asymptotic performance compared to several leading baselines on a set of challenging image-based Mujoco control tasks.
    * ROTOC shows significant robustness to observational distractors, including Gaussian white noise and dynamically changing natural video backgrounds, often outperforming baselines in these settings.
4.  **Thorough Ablation Studies and Analysis**:
    * Extensive ablation studies systematically validate the contributions of different components: chunk-wise mini-batching, the individual lower bounds (KL and InfoNCE), and the intrinsic reward.
    * The paper provides evidence that chunk-wise mini-batching increases multi-step predictability of the learned representations, linking this to improved performance.
    * Visualizations (e.g., t-SNE plots) suggest that ROTOC learns representations that effectively capture task-relevant features (like robot configurations) while being invariant to background distractors.


## Weaknesses

1.  **Approximation of the Core Objective**:
    * While the ROTOC objective $\max \mathcal{C}(z_1; \dots; z_T)$ targets long-term correlation, the primary tractable lower bound decomposes into a sum of per-step mutual information terms $\sum I(z_{t+1}; z_t, a_t)$. As the authors acknowledge, this decomposition might inherently lose some of the direct incentive to capture long-term correlations beyond single steps.
    * Chunk-wise mini-batching is a practical and empirically effective technique to mitigate this, but it's an indirect way to optimize the original global objective, as the loss function itself remains a sum of per-step terms.
2.  **Potential for More Direct Multi-Step Modeling**:
    * The authors themselves suggest that "it would also be interesting to explore objectives that better capture multi-step correlations based on multi-step prediction models". This indicates an area where the current approach could be extended for even stronger long-term dependency modeling, moving beyond the current one-step latent transition model $q_{\psi}(z_{t+1}|z_t, a_t)$.
3.  **Clarity on Negative Sample Selection for InfoNCE**:
    * For the InfoNCE lower bound, negative samples $z_{t+1}^*$ are drawn from the marginal distribution $p(z_{t+1})$. While standard, practical implementation often involves sampling negatives from the current batch or a memory bank. Further details on how these negative samples are specifically obtained/approximated in the experiments could enhance reproducibility

---

> ### Author Response · Authors · 2025-06-02
> **Rebuttal by Authors**
>
> Thank you for providing valuable comments, and appreciating our contributions, and thorough experimentations.
>
> > Clarification of the Link Between Per-Step Loss and Global ROTOC Objective with Chunk-Wise Mini-Batching
>
> Our motivation for chunk-wise mini-batching precisely aligns with your latter argument: We expect the aggregated gradients to better reflect the dependencies across the entire rollout. We could probably also motivate it as a technique to encourage smoother representation manifolds, but we regard this rather as a consequence of better aggregated gradients that are less likely to point towards bad local optima. We agree that we should elaborate on our motivation, and changed Section 4.5.1 as follows:
>
> The loss function (Eq. 7) is optimized using stochastic gradient descent based on mini-batches sampled from the replay buffer of the agent. When sampling individual transitions independently, we would only use a single transition for many trajectories, which may prevent us from capturing longer-term correlations. Instead, we propose to sample a mini-batch of sequence chunks $(s_t, a_t, s_{t+1})_{t=k}^{L+k-1}$ with chunk length $L$ from the replay buffer. While our loss (Eq. 7) decomposes additively across time steps, implying that chunk-wise sampling does not alter the optimized loss, it intuitively biases the optimization toward local optima with better long-term consistency by ensuring the mini-batch contains subsequent time steps. This is because gradients from independent single-step transitions can steer the model into local optima exhibiting only local temporal consistency, suboptimal for multi-step prediction. Even if subsequent time steps are eventually sampled in other minibatches, escaping such local optima can be difficult. Instead, chunk-wise mini-batching ensures the aggregated gradient of subsequent transitions will agree on pushing towards a globally consistent representation, effectively canceling out signals that might otherwise lead to merely locally consistent solutions.
>
> > Explicit Details on InfoNCE Negative Sample Strategy
>
> Sorry for this lack of clarity. In practical implementation, we randomly sample a minibatch of (sequential) transitions $(s_t, a_t, s_{t+1})$, and obtain a minibatch of positive samples $(z_t, a_t, z_{t+1})$ by encoding the minibatch using our models. For a given positive sample, the negative sample set $z^*_{t+1}$ is constructed by using all other embeddings $z_{t+1}$ of the same mini-batch. In our revision, we provide more details about the implementation of negative samples in Appendix B.
>
> > Brief Acknowledgment of Hyperparameter Tuning Scope for Representation Learning Components
>
> Thank you for pointing out this problem. We now add a sentence to the Experimental Setup section to explain how we choose the key hyperparameters for ROTOC: "We select the coefficients $\alpha$ and $\lambda$, and the chunk length $L$ by performing hyperparameter tuning on the standard Cartpole Swingup task and subsequently fix them for all other tasks."

---

> ### Comment · Action_Editor_7PNL · 2025-06-12
>
> Thank you for all your efforts in reviewing this paper so far.  Have you find the authors' rebuttal satisfactory? Is there any remaining issues? If you need to discuss anything only among reviewers, there is a separate Discussion thread. At the very least, please acknowledge the authors' response..
>
> Thank you,
> Action Editor

---

### Author Response · Authors · 2025-06-02
**General rebuttal by Authors**

Thank all the reviewers for carefully reviewing our submission, and providing constructive feedback.

We updated the manuscript based on the comments, and highlighted all the changes in blue. We summarized the changes as follows:

- Clarifying the effect of chunk-wise mini-batching sampling on improving long-term correlations (Sec. 4.5.1, for Reviewer 7GyU).

- Providing more details on obtaining InfoNCE negative samples (Appendix B.6, for Reviewer 7GyU).

- Adding explanations about the choice of some key hyperparameters of ROTOC (Sec. 5.1, for Reviewer 7GyU).

- Providing more discussions about the differences to the related work, RPC. (Sec. 2, for Reviewer dHTV)

- Improving the implementations of RPC and retesting its performance on all standard, noisy, and natural MuJoCo tasks (Sec. 5.2.1, Sec. 5.2.2 and Appendix C, for Reviewer dHTV).

- Adding evaluations on computational complexity for ROTOC and other approaches (Appendix E, for DXqP)

In addition, we added a discussion about the differences to the work [1], which recently introduced total correlation to reinforcement learning. (Sec. 2)

[1] Bang You, Puze Liu, Huaping Liu, Jan Peters, Oleg Arenz. Maximum Total Correlation Reinforcement Learning, https://arxiv.org/abs/2505.16734

---

### Comment · Editors_In_Chief · 2025-08-27

On August 27, by request of the authors, the EiCs uploaded a new camera-ready PDF for the paper. The updated version now marks one of the authors as the corresponding author.

---

### Decision · Action_Editor_7PNL · 2025-07-27

**Recommendation:** Accept with minor revision

**Additional Comments:**

## Summary ##
The paper proposes maximizing the Total Correlation (TC) of the rollout trajectory in order to learn good representations that improves the temporal consistency for reinforcement learning agents. The TC of multiple random variables is a measure of dependency between them, so by maximizing the TC of the embedding of the whole trajectory, the method attempts to learn a representation with high temporal consistency. The paper approximates TC into step-wise mutual information terms (Eq. 4), which itself is then lower bounded using variational approximations (Eq. 5 and 6). The resulted method performs well in comparison with several other methods.


## Evaluation ##
All three reviewers are positive about the paper (two Leaning Accept and one Accept). They believe that the Rollout Total Correlation (ROTOC) is a novel objective (Reviewers 7GyU and DXqP) and the empirical evaluation is comprehensive (all reviewers).
On the critical side, they have mentioned issues such as

- indirect optimization of the TC (7GyU, dHTV),
- the chunk-wise mini-batching is a heuristic workaround rather than a direct optimization of TC (7GyU),
- closeness of the proposed lower bound in approximating TC is not established (DXqP).


I read the paper myself, before closely reading the reviews, to form an independent opinion. I share my comments below. In general, I agree with the positive stance of reviewers and find the contributions promising. I also see a few unclear conceptual and methodological issues that can be better addressed or at least acknowledged in a revised version of the paper. Overall, I believe even though the paper may not fully justify all its design choices from first principles, the overall performance of the method is strong and the ideas are novel and potentially impactful. I believe it is a worthwhile contribution to the community. Therefore, I recommend acceptance of the paper with minor revisions, with the hope that the method will be understood better in the future.

## My comments ##

**Suitability of TC for representation learning**

It is not clear why the Total Correlation is actually a good objective for representation learning. There are comments such as "the energy of an approximately closed system is a a very powerful feature, precisely due to its temporal consistency, which allows for long-term predictions" (bottom of page 1), which is supposed to justify the importance of temporal consistency, and hence TC as the quantity to maximize. The authors provide similar argument in answering Reviewer dHTV, though at the end admit that " ... it is indeed not obvious why exactly they should help in a model-free RL setting".

To investigate this further, I'd like to note that TC is maximized when one random variable (for example, z1) determines the rest of the random variables. If I am not mistaken, and please correct me if I am, this means that if the representation collapses to a single point, the objective in Eq. 1 is maximized. A collapsed representation is certainly not a good representation. This suggests that this objective may not be an ideal representation learning objective on its own.

Of course, there are several possible answers to it:

1) maybe the later approximations made in the paper prevent this to happen
2) maybe because this is not the only objective being optimized, but the combination of this objective and the SAC objective are optimized, the learned representation is a good one (though I am not certain whether that SAC loss propagates back to representation layers or not)
3) maybe the dynamics of the optimizer avoid finding the optimal solution of TC (or approximation thereof), similar to results available for BYOL (see Tang et al., "Understanding Self-Predictive Learning for Reinforcement Learning," ICML, 2023) and other representation learning methods in RL (see Voelcker, Kastner, Gilitschenski, Farahmand, "When does Self-Prediction help? Understanding Auxiliary Tasks in Reinforcement Learning," RLC, 2024 [disclaimer: I am a co-author of this paper. I am not requesting the authors to cite it, but it is very relevant for their research.]).


**Step-wise lower bounding**

Ignoring my previous comment regarding the suitability of TC for the moment: I am worried that lower bounding in Eq. 4, which reduces the objective to the sum of mutual informations of each step, loses the whole point of maintaining the temporal consistency, whose importance is the main motivation of the paper. If so, is it justified to call the paper "Rollout Total Correlation ..." instead of "Approximate Rollout Total Correlation ..."

**Chunking**

I realize that the chunk-wise mini-batching is supposed to partially address the previous concern. I do not find the justification of the revised Section 4.5.1 very convincing though: "While our loss (Eq. 7) decomposes additively across time steps, implying that chunk-wise sampling does not alter the optimized loss, it intuitively biases the optimization toward local optima with better long-term consistency by ensuring the mini-batch contains subsequent time steps."

Can the authors extend their paper by studying the effect of chunking on the optimization process (and not the final performance, which is already done)?
For example, maybe one can empirically compute the autocorrelation of the gradient as a function of how close two data points are within a rollout sequence. And also compute the variance of the gradient of the loss with varying sizes of chunking.
Or any other meaningful empirical measure that investigates the effect of chunking on the dynamics of the optimizer, as opposed to its effect on the final performance (for which there are already some results).

These experiments may address the hypotheses in Section 5.3.1 of the paper ("We hypothesize that a large chunk length either ..." and "we hypothesize that too large chunk lengths lead to too small variance in the stochastic gradient estimates, which may hurt the performance.").

**Relation to bisimulation**

I was surprised not to see any mention of bisimulation metric as a way to learn representation. Some key papers are:

- Ferns, Panangaden, and Precup, "Metrics for finite Markov decision processes," UAI, 2004.
- Castro, "Scalable methods for computing state similarity in deterministic Markov decision processes," AAAI, 2020.
- Zhang, McAllister, Calandra, Gal, and Levine, "Learning invariant representations for reinforcement learning without reconstruction," ICLR, 2021. [This paper is actually cited, but as a reference for the experimental setup, and not its core contribution in approximate bisimulation metric learning for representation learning.]
- Kemertas and Jepson, "Approximate Policy Iteration with Bisimulation Metrics," TMLR, 2022. [Disclaimer: Kemertas is my PhD student, but I was not directly involved in this paper.]

**Inequality in Equation (5)**

The right hand side of Equation (5) is negative (more precisely, non-positive), as the KL divergence is always non-negative and there is a negative sign there. Since the mutual information (left hand side) is always positive (more precisely, non-negative), this inequality is always satisfied. Maximizing the RHS then at best leads to the RHS being zero, which does not put any pressure to maximize the LHS, which is positive anyway. What is being enforced here?
There is a statement just after the equation, stating "Minimizing the bound encourages ...", which makes me confused, as the minimizer of the RHS leads to a large negative number. Is there any typo here?


**Other Comments**

- Would you please clarify whether the updates of encoders, actor and critic, etc. in Algorithm 1 are done separately, or are the gradients added to each other?

- In the References, "Chenjia Bai et al." doesn't show the full list of authors. There is at least another example like this among the references.

**Audience:**

Yes

**Audience Explanation:**

Yes, this is very relevant to researchers interested in reinforcement learning.

**Claims And Evidence:**

Yes

**Claims Explanation:**

Yes, the claims are mostly justified. Read my full comments for more detail.

---

> ### Author Response · Authors · 2025-08-25
> **Reply by Authors**
>
> Thank you very much for handling the review process and providing constructive independent reviews. We highly appreciate that effort and took the feedback very carefully into account.
>
> > TC is maximized when the representations collapses to a single point
>
> Thank you for this valuable feedback. The total correlation is defined as the KL divergence between the joint distribution of all embeddings and actions and the product of their marginals. If all representations and actions are degraded to a single constant, the KL divergence becomes zero, where the total correlation is minimized instead of maximized. Hence, maximizing the total correlation avoids the collapsed representation. Intuitively, the total correlation can be understood as the extra information required to encode each random variable independently, compared to using an optimal joint code for the entire trajectory. If the uncertainty of the whole trajectory is minimized and meanwhile the uncertainty of each random variable is maximized, the total correlation is maximized.
>
> > Step-wise lower bounding
>
> One of our contributions is to introduce the rollout total correlation as a novel objective for representation learning in image-based reinforcement learning. While our practical implementation only approximately maximizes it by using a step-wise lower bound, we want to emphasize our focus on this underlying objective.
>
> > Studying the effect of chunking on the optimization
>
> Thank you for your constructive suggestion. To investigate the effect of chunking on the optimization, in our revision we now evaluate the variance of the gradients of the parameters of the encoder with respect to the loss over different time steps within trajectories. Specifically, we collected 20 independent evaluation trajectories, computed the gradients of the parameters of the encoder every time step within trajectories, and calculated the variance of gradients across all time steps. As the same loss function and test data are used, the computed metric only depends on the model parameters that were learned with the different chunk lengths. Hence, the metric is used for examining the local optima (in terms of gradient variance along the same trajectories) that are reached when using different chunk lengths during training.  The results support our hypothesis that chunk-wise mini-batching yields more consistent gradients along a given trajectory. This trend is particularly evident when increasing the number of samples in the batch, where the gradient variance significantly increases without chunking, but remains approximately constant, when using chunk-wise mini-batching. We added these results in Appendix D.5 of the revision.
>
> > Relation to bisimulation
>
> Thank you for bringing these bisimulation-based methods to our attention. We now add discussion to these approaches at the beginning of the Related Work Section in our revision.
>
> > Inequality in Equation (5)
>
> The lower bound in the RHS (the negative KL divergence) is always non-positive and too loose for estimating the mutual information. However, as we aim to learn consistent representations, maximizing the lower bound is still meaningful, since minimizing the KL divergence between the encoder and the transition model can encourage the encoder to learn temporally predictable embeddings of states. Our empirical results in Table D.2 demonstrate that maximizing the lower bound improves the predictability of embeddings.
>
> > Is there any typo here?
>
> Indeed, it is a typo. We now replace the "Minimizing" with "Maximizing".
>
> > Clarification on the updates of encoders, actor and critic
>
> Sorry for the lack of clarity.  In Section 4.7, we now clarify that we let the gradients of the critic backpropagate through the online deterministic encoder, since the environment reward can provide useful task-relevant information. Following previous works, we stop the gradient of the actor through the embedding, since this can degrade performance by implicitly modifying the Q-function during the actor update.
>
> > Authors presentation errors
>
> We carefully checked the references and have fixed several author errors in our revision.